# On the Interdependence between Data Selection and Architecture Optimization in Deep Active Learning

**Pradeep Bajracharya** *pb8294@rit.edu*
*College of Computing and Information Sciences*
*Rochester Institute of Technology*

**Rui Li** *rxlics@rit.edu*
*College of Computing and Information Sciences*
*Rochester Institute of Technology*

**Linwei Wang** *linwei.wang@rit.edu*
*College of Computing and Information Sciences*
*Rochester Institute of Technology*

**Reviewed on OpenReview:** *https: // openreview. net/ forum? id= dHcoMrmWcE*

## Abstract

Deep active learning (DAL) studies the optimal selection of labeled data for training deep neural networks (DNNs). While data selection in traditional active learning is mostly optimized for given features, in DNN these features are learned and change with the learning process as well as the choices of DNN architectures. How is the optimal selection of data affected by this change is not well understood in DAL. To shed light on this question, we present the first systematic investigation on: 1) the relative performance of representative modern DAL data selection strategies, as the architecture types and sizes change in the underlying DNN architecture (Focus 1), and 2) the effect of optimizing the DNN architecture of a DNN on DAL (Focus 2). The results suggest that the change in the DNN architecture significantly influences and outweighs the benefits of data selection in DAL. These results cautions the community in generalizing DAL findings obtained on specific architectures, while suggesting the importance to optimize the DNN architecture in order to maximize the effect of active data selection in DAL.

## 1 Introduction

Active learning (AL) is a long standing research area that studies how to carefully select the most informative samples to label in order to best improve learning given a limited labeling budget (Cohn et al., 1996; Settles, 2009). As the success of deep neural networks (DNNs) continues, there have seen substantial developments in bringing the success of AL to DNNs, creating a fast-growing subarea known as deep active learning (DAL) (Ren et al., 2021). In its essence, DAL starts with training a DNN on an initial pool of labelled data, followed by optimizing an *acquisition function* to select new data to be labelled to improve the DNN. This iteration continues until a labeling budget or desired accuracy is achieved (Ren et al., 2021).

The main progress in DAL has been revolving around the design and developments of acquisition functions to improve the selection of training data. Most data acquisition strategies can be categorized into efforts that exploit DNN uncertainties (Ranganathan et al., 2017; Wang et al., 2016; Li et al., 2017; He et al., 2019; Ostapuk et al., 2019; Gal et al., 2017b; Freytag et al., 2014; Käding et al., 2016; Yoo & Kweon, 2019; Huang et al., 2022), explore the diversity of unlabeled data (Wang et al., 2017; Sener & Savarese, 2017; Geifman & El-Yaniv, 2017; Shui et al., 2020; Zhang et al., 2020b; Sinha et al., 2019; Kim et al., 2021), or combine the advantages of the above two in a hybrid fashion (Liu et al., 2016; Coletta et al., 2019; Zhdanov, 2019; Ash et al., 2020; Shui et al., 2020; Kong et al., 2022; Wang et al., 2022; Settles et al., 2007; Shukla, 2022).

Additional approaches also consider the generation of informative examples (Mahapatra et al., 2018; Mayer & Timofte, 2020; Zhu & Bento, 2017), utilizing the unlabeled data to pretrain the DNN feature extractor (Siméoni et al., 2021) or in a semi-supervised fashion during DAL (Siméoni et al., 2021; Gao et al., 2020), or exploiting the training dynamics of the DNN (Wang et al., 2022).

In this paper, we bring to the research community an under-explored yet important question in DAL: the interdependence between choice of DNN architecture on data optimization during active learning. Traditional AL developments are typically associated with pre-defined features, where data selection is primarily optimized for task decision boundaries on the given feature space. In contrast, for DNNs, the feature space is unknown and is learned as training progresses. How does this affect optimal data selection strategies? This raises questions about the potential interdependence between DNN architectures and data acquisition strategies that are not yet well understood.

First, it is not clear how DAL evaluation may be dependent on (or agnostic to) the choices of DNN architectures, considering that the vast majority of DAL works are based on predefined and fixed DNN architectures (Geifman & El-Yaniv, 2018). Indeed, several recent works have investigated the reproducibility of DAL evaluations (Beck et al., 2021; Munjal et al., 2020; Mittal et al., 2019), noting that performance gaps among various DAL methods are inconsistent across experimental settings. Some also noted that the gain of DAL over random acquisition is in general marginal compared to the use of other regularization techniques (Beck et al., 2021; Munjal et al., 2020). These studies however have not focused on the effect of DNN architecture choice. In Focus 1 of this paper, we address this knowledge gap by systematically evaluating the relative performances of seven representative DAL acquisition functions for a given DNN architecture, across two convolution-based DNN architecture types on image datasets and three transformer-based DNN architecture types on text datasets, each with different sizes.

Second, it is not clear how DAL developments can be improved by DNN architectural optimization, or what may be the most effective approach to combine DNN training data optimization with architecture optimization. Until now, only one previous work examined the effect of incremental DNN architectural search as DAL proceeds, showing that architecture optimization brought improvements compared to the use of fixed architectures during DAL (Geifman & El-Yaniv, 2017). It however was limited to the investigation of a particular architecture optimization approach, designed for a particular type of DNN architectures (RESNET-18). Nor did it have a focus on the relative importance between optimizing the DNN architecture *versus* optimizing data selection. In parallel, there have been increasing DAL works that put an explicit focus on DNN feature space, mostly done via pre-training a feature extractor using unlabeled data (Siméoni et al., 2021) and/or simultaneously improving the learned feature during DAL using unlabeled data in addition to the acquired labeled data (Kong et al., 2022; Hacohen et al., 2022; Wang et al., 2022). All of these works, however, are performed with fixed DNN architectures. In Focus 2 of this paper, we address this knowledge gap by systematically evaluating how architecture optimization of the DNN feature extractor may affect DAL, considering three possible approaches: 1) jointly optimizing the DNN architecture with the acquired labeled data throughout the course of DAL, or 2) pre-optimizing the DNN architecture using the unlabeled data prior to DAL, or 3) pre-optimizing the DNN architecture using the initial label data prior to DAL. We conduct this investigation with three representative approaches to DNN architecture optimization that allow manageable computation when used repetitively over the course of DAL: two Bayesian architecture inference methods (KC et al., 2021; Lee et al., 2018) and one neural architecture search (NAS) method (Chen et al., 2019), all in comparison to their fixed counterparts on the same spectrum of DAL strategies considered in Focus 1.

Experimental results from Focus 1 demonstrated that the relative performance of DAL acquisition functions was substantially influenced by the underlying DNN architecture, cautioning the community in generalizing DAL findings obtained on specific architectures. Results from Focus 2 showed that optimizing DNN architecture substantially benefited DAL. More interestingly, the gain of performance induced by an optimized *vs.* pre-defined DNN architecture appeared to significantly *outweighed* the choice of DAL acquisition strategies. This leaves a potentially important implication for DAL research: while the importance of DNN architecture choices is widely accepted in standard "passive" learning, it may be especially crucial and may suggest an under-explored research avenue in DAL in order to maximize the effect of active data selection when the labeling budget is limited.

## 2 Related Works

**Deep active learning (DAL):** DAL research has flourished over the years with the design of various strategies to select data from the unlabelled pool. The strategies could be broadly divided intro three categories. Uncertainty-based strategies seek examples that a DNN is most uncertain about. A variety of measures has been proposed to represent this broadly-defined uncertainty, including entropy (Joshi et al., 2009), BALD (Houlsby et al., 2011; Shelmanov et al., 2021), least confidence based on softmax outputs (Settles, 2009), margin sampling (Scheffer et al., 2001), expected gradient length (Settles et al., 2007; Huang et al., 2016; Zhang et al., 2017; Shukla, 2022), changes in outputs in response to input perturbation(Freytag et al., 2014; Käding et al., 2016), and estimation of DNN loss (Yoo & Kweon, 2019; Huang et al., 2022). Diversity-based strategies seek samples that are representative of the unlabelled data using approaches such as density clustering (Wang et al., 2017), coreset optimization (Sener & Savarese, 2017; Geifman & El-Yaniv, 2017), and leveraging adversarial networks (Zhang et al., 2020b; Sinha et al., 2019; Kim et al., 2021). Hybrid strategies combine these two approaches to sample diverse data which the DNN is most uncertain about (Ash et al., 2020; Shui et al., 2020; Wang et al., 2022; Kong et al., 2022), such as by considering the magnitude as well as diversity of DNN gradients (Settles et al., 2007; Shukla, 2022).

In addition to these pool-based active learning where new training data is obtained by querying an unlabelled pool, there are generative approaches (Zhang et al., 2020b) that generate examples informative to the current model. These approaches leverage generative adversarial network (GAN) to generate informative data examples that has high entropy (Mayer & Timofte, 2020) or are closer to the decision boundary (Zhu & Bento, 2017) (Mahapatra et al., 2018; Mayer & Timofte, 2020; Zhu & Bento, 2017). Additional approaches include utilizing the unlabeled data to pretrain the DNN feature before DAL (Siméoni et al., 2021) or using it in a semi-supervised fashion during DAL (Siméoni et al., 2021; Gao et al., 2020), as well as exploring DNN training dynamics (Wang et al., 2022) as measured by the derivative of training loss with respect to the number of iterations assuming that models training faster generalize better. These developments are seen in both image as well as text data domains.

Most of these existing works were conducted on specific choices of DNN architectures, such as MLPs (Ash et al., 2020), LeNet (Geifman & El-Yaniv, 2017; Hu et al., 2021), CNNs (Gal et al., 2017b), and different versions of VGG and RESNET (Ash et al., 2020; Shui et al., 2020) on image data, or BERT (Zhang et al., 2020a; Schröder et al., 2021; Wertz et al., 2022) and its two variants – DistilBERT (Schröder et al., 2021; Kirk et al., 2022) and RoBERTa (Lu & MacNamee, 2020) on text data. A lack of consistency regarding the choices of DNN architectures exist across existing studies, and it is not clear how the reported DAL evaluations may be dependent on (or agnostic to) the choices of DNN feature extractors, a critical question that will be systematically investigated in this paper.

**Systematic evaluation of DAL methods:** An observation emerging in recent works (Mittal et al., 2019; Munjal et al., 2020; Beck et al., 2021) is the inconsistency and reproducibility of the relative performance of DAL methods across experimental settings. The lack of unified experimental setting, such as size of the initial labeled pool, acquisition size, total labeling budget, random seeds, batch size, and optimizers have been credited for the inconsistencies of results reported (Munjal et al., 2020; Beck et al., 2021). It was further shown that the gain of DAL over random acquisition is in general marginal compared to other strategies, such as network regularization, data augmentation, and semi-supervised techniques (Munjal et al., 2020; Beck et al., 2021). This paper will add to these findings focusing on the effect of optimizing the architecture of DNN feature extractor on DAL.

**DNN architecture optimization in DAL:** There is a large body of literature in deterministic optimization or Bayesian inference of DNN architectures (Zoph & Le, 2016; Zoph et al., 2018; Kasim et al., 2020; Feng & Darrell, 2015; Lee et al., 2018; Dikov & Bayer, 2019; KC et al., 2021), supporting the notion that the complexity of DNN feature extractors has substantial impact when *passively* learning from given data. To date, only one work investigated the effect of optimizing DNN architecture in the context of active data selection as DAL proceeds (Geifman & El-Yaniv, 2018). Specifically, an incremental architectural search method was formulated over a modularly reduced search space customized for RESNET-18, integrated and evaluated with three existing DAL data acquisition strategies. This paper will substantially expand the scope

of this previous study by: 1) investigating both supervised joint training as well as unsupervised pre-training as alternative approaches to combine DNN architecture and data optimization, 2) including a variety of DNN architectures especially convolutional architectures (VGG- and RESNET-variants) for image data and transformer architectures (BERT, DistilBERT, and RoBERTa) for text data, 3) including three architecture optimization methods representative of both Bayesian architecture inference (KC et al., 2021; Lee et al., 2018) and deterministic NAS (Chen et al., 2019) approaches, and eventually 4) deriving insights into the relationship between DNN architecture optimization and data optimization during DAL.

## 3 Focus 1: The Effect of DNN Architectures on DAL Performances

### 3.1 Methodology

In Focus 1, we systematically investigate whether and how the *relative performance* of existing DAL acquisition strategies may depend on the underlying DNN feature extractor, especially its difference in choices of architecture types and sizes. We consider classification tasks on both image and text data. On image tasks, we consider two convolutional DNN architecture types (RESNET and VGG) that are mostly commonly used in DAL literature and each with three different sizes (RESNET-18/34/50 and VGG-11/16/19). On text tasks, we consider three transformer architectures (BERT, DistilBERT, and RoBERTa) that are most commonly used in text-based active learning tasks. We consider seven state-of-the-art DAL acquisition functions representative of uncertainty-based (Settles et al., 2007; Scheffer et al., 2001; Joshi et al., 2009; Houlsby et al., 2011), diversity-based (Sener & Savarese, 2017; Ash et al., 2020), and hybrid strategies (Ash et al., 2020), each in combination with or without unsupervised pretraining as the seventh DAL strategy presented in (Siméoni et al., 2021).

As our main focus is to understand how the performance gap among various acquisition strategies may be affected by the choice of DNN architectures, we devise metrics to measure the *relative performance* of different acquisition strategies, and examine its changes across choices of DNN architectures. We further examine how such effects change as the data selection is affected by the size of the initial labeled pool, the size of each acquisition, and the presence of data augmentation. We leave out investigations on other DNN hyperparameters or regularization techniques that have been studied in previous works (Munjal et al., 2020; Beck et al., 2021).

**DAL acquisition strategies:** We consider the following acquisition functions, against random sampling as the baseline. For all below, we consider $L$ as the labeled datasets, $U$ as the unlabeled pool of dataset and $D$ as the complete dataset i.e. $D = L \cup U$. These are further explained in Section 3.2.

1. *Least confidence sampling* (Settles et al., 2007) chooses instances ($x^*$) with the least probability scores of the predicted class $p(c)$ among class $c \in C = \{1, 2, ..., k\}$ in the output for a given unlabelled data $x_i$ i.e. $x^* = argmin_{x_i \in U} p(c|x_i)$.

2. *Margin sampling* (Scheffer et al., 2001) selects instances ($x^*$) with the smallest difference between the first and second largest class label probability, assuming that a confident model is characterized by a substantial gap between the predicted label probability and the second-highest label probability:

$$x^* = argmin_{x_i \in U}[p(y = c_1|x_i) - p(y = c_2|x_i)] \tag{1}$$

3. *Entropy sampling* (Joshi et al., 2009) selects instances ($x^*$) a DNN is most uncertain about as measured by the entropy ($H$) calculated from the output softmax probabilities:

$$H = -\sum_{c \in C} p_c * log(p_c)$$
$$x^* = argmax_{x_i \in U} H(x_i) \tag{2}$$

4. *BALD sampling* (Houlsby et al., 2011) selects instances ($x$*) that generates disagreeing predictions that the model is the most uncertain on average information gain ($I$) (Gal et al., 2017b) i.e.

$$
\begin{aligned}
I(y; w|x, L) &= H(y|x, L) - E_{p(w|L)}[H(y|x, w, L)] \\
x* &= argmax_{x_i \in U} I(y; w|x_i, L)
\end{aligned}
\tag{3}
$$

5. *Coreset Sampling* (Sener & Savarese, 2017) selects unlabeled instances that are the most different from existing labelled sample based on their feature distances:

$$
x* = argmax_{i \in U} \ min_{j \in L} \triangle (x_i, x_j)
\tag{4}
$$

6. *BADGE sampling* (Ash et al., 2020) selects instances that generates diverse but also high gradient magnitudes in the penultimate layer of the DNN.

7. *Unsupervised Pretraining*: Based on Simeoni et al. (2021), we adopt a two-step pretraining strategy for both image and text datasets. This pretraining involves a combination of alternate unsupervised clustering task and classification task supervised by the clustering labels. We begin with random initialization of the network parameters and the features from the penultimate layers are clustered using k-means clustering. These generated pseudo- labels are then utilized as the ground truth for a subsequent supervised classification task which in turn updates the network parameters. The networks, once fully trained, serve as the initial models for all subsequent active learning experiments. This strategy is used in combination with all acquisition functions described above.

These acquisition functions are the commonly used benchmarks in DAL research. Among them, the first four are representative of uncertainty-based acquisition strategies, with Entropy and BALD calculated based on Bayesian drop-out strategies (Gal et al., 2017b). Coreset is representative of diversity-based strategies, and BADGE is a representative hybrid strategy. Each of these acquisition functions are tested without and with unsupervised pretraining as the seventh DAL strategy considered.

**Network architectures:** On image data, we consider two convolutional DNN architecture types that are mostly used in DAL literature, each with three different sizes: VGG-11, -16, -19; (Simonyan & Zisserman, 2014), and RESNET-18, -34, -50 (He et al., 2016). On text data, we consider three transformer architectures namely BERT (Devlin et al., 2018), RoBERTa (Liu et al., 2019) and DistilBERT (Sanh et al., 2019) due to their prominent use in text based active learning tasks. Details of DNNs are described in the Appendix B.

**Evaluation metrics:** We consider two quantitative metrics: 1) labeling efficiency as described in (Beck et al., 2021), which measures the amount of data required in comparison to random acquisition (as a ratio) to achieve the same test accuracy; and 2) a new metric that measures the percentage of gain in test-accuracy over random acquisition at each acquisition round averaged over all acquisition rounds. We use these two metrics to compare the relative performance of the considered acquisition functions across DNN architecture types and sizes.

## 3.2 Experiments and Results

Experiments in Focus 1 were performed on four image datasets including MNIST (Deng, 2012), Fashion MNIST (Xiao et al., 2017), CIFAR10 (Krizhevsky, 2009b), and SVHN (Netzer et al., 2011), and four text datasets including AGNEWS (Zhang et al., 2015), Banks77 (Casanueva et al., 2020), DBPedia (Auer et al., 2007) and QNLI (Wang et al., 2018). Let $D$ be the complete dataset divided into initially labelled data $L = \{x_l, y_l\}_{l=1}^{|L|}$ and unlabelled pool of data $U = \{x_u\}_{u=1}^{|D|-|L|}$ of size $|L|$ and $|D| - |L|$, respectively. We initialized the networks on the labelled set $L$. In each acquisition round, the acquisition function of choice was used to choose $|A|$ number of data points from $U$ to be labelled next. The newly labelled data was added back to the labelled set $L$ to retrain the network and the process was repeated until $|B|$ number of unlabelled data has been labelled.

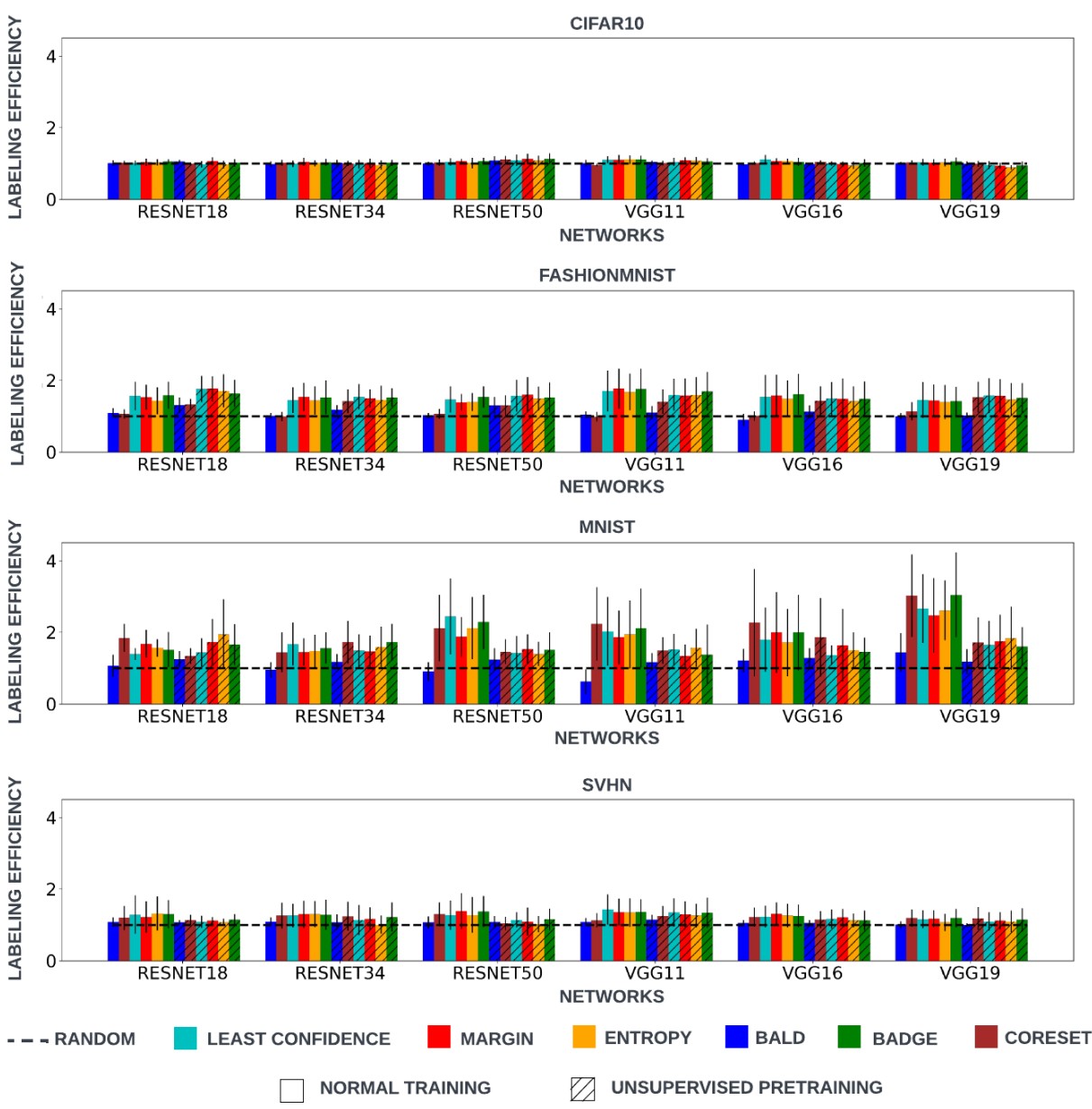

Figure 1: Comparison of labeling efficiency over random acquisition, for all six acquisition function with (hatched) and without unsupervised pretraining across four datasets and all DNN architectures on image classification tasks. The dashed horizontal line shows reference of random acquisition. Results show that the relative performance of acquisition strategies varies both across DNN choices on the same dataset, as well as across datasets.

On image dataset, for consistent experimental settings, we used an initial balanced labelled data size of 1000 unless otherwise stated. The remaining data were divided into 90-10% training-validation split and the size of the unlabelled pool was 30,000. A total of 25 acquisition rounds were performed with an acquisition size of 1000. Each round is set to train for 500 epochs with early stopping added when the training accuracy reached 99% or validation accuracy remained unchanged for 50 epochs. On text data, we used an initial labelled data size of 100 with 25 rounds of acquisition rounds performed with an acquisition size of 100. The experiments are run for 5 epochs following (Devlin et al., 2018). All experiments were run for three seeds on

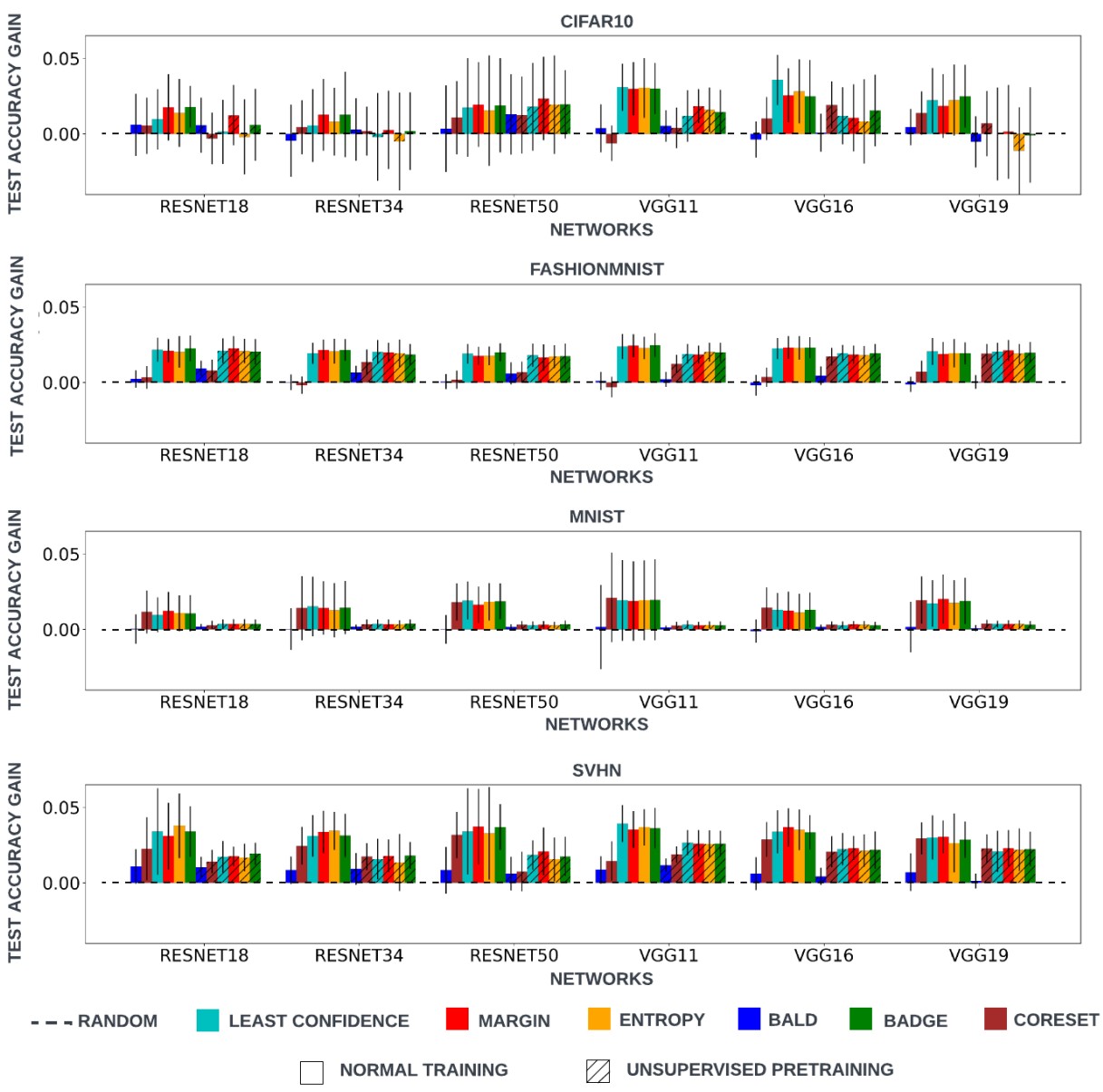

Figure 2: Comparison of gain in test-accuracy over random acquisition (as 0.00 test accuracy gain), for all six acquisition function with (hatched) and without unsupervised pretraining across four datasets and all DNN architectures on image classification tasks. Similar to Figure 1, results show that the relative performance of acquisition strategies varies both across DNN choices on the same dataset, as well as across datasets.

workstations with RTX 2080Ti GPU and 32 GB of RAM as well as P8 and V100 GPU provided by Research Computing at Rochester Institute of Technology (Rochester Institute of Technology, 2024). We used the DISTIL github repo (Dani et al., 2021) as a base skeleton for our image experiments and dal-toolbox github repo (Rauch et al., 2023) for text experiments. We used dbViz github repo (Somepali et al., 2022) to visualize the decision boundaries images across different networks and acquisition functions for image dataset.

**Relative performance of acquisition functions:** Figure 1 summarizes the labeling efficiency and Figure 2 summarizes the gain of DAL over random acquisition for all network architectures on the image datasets. Figure 3 – Figure 4 summarize similar results on the text datasets. The numerical values for

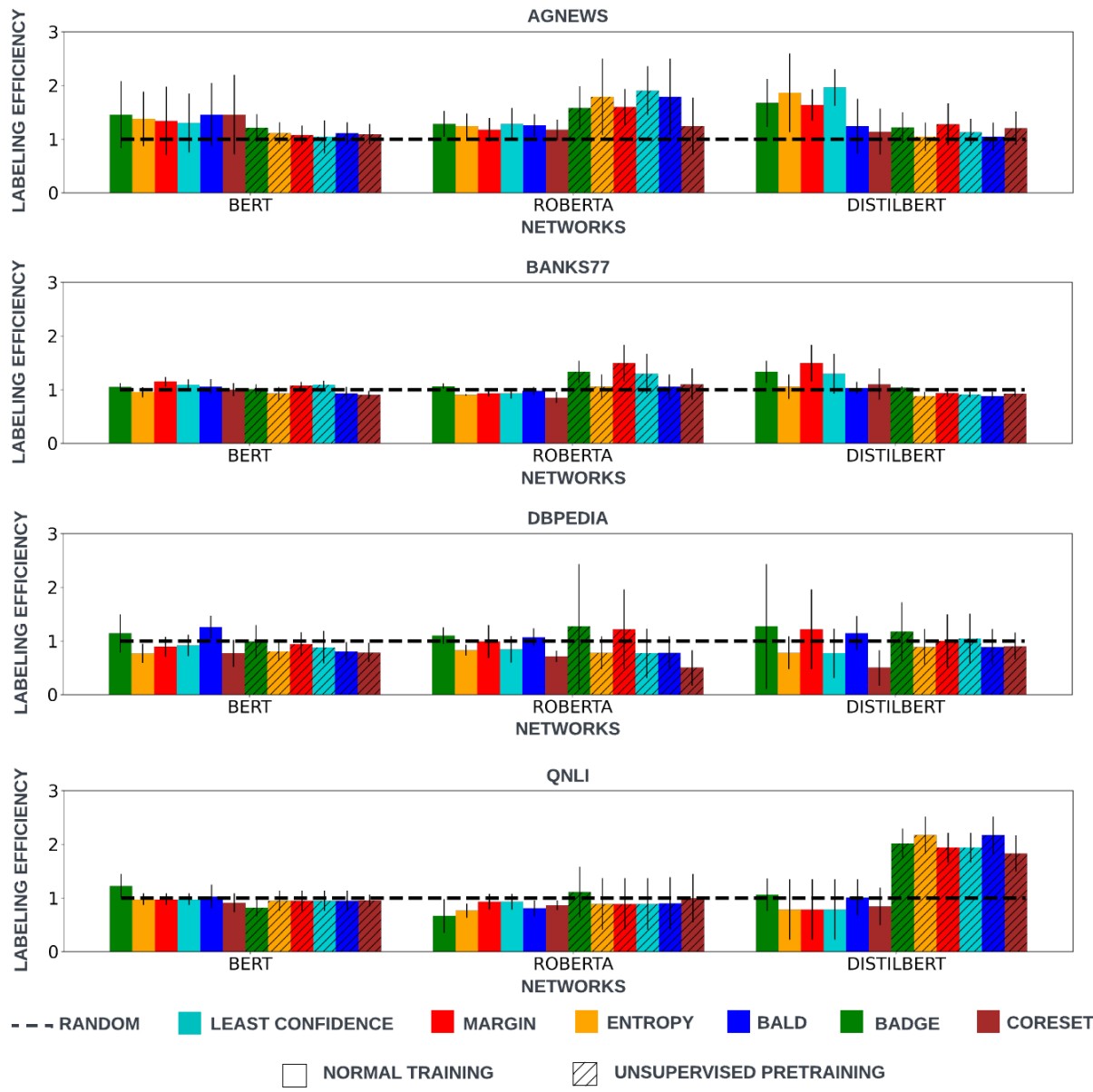

Figure 3: Comparison of labeling efficiency over random acquisition, for all six acquisition function with (hatched) and without unsupervised pretraining across four datasets and all transformer architectures on text tasks. The dashed horizontal line shows reference of random acquisition. Results show that the relative performance of acquisition strategies varies both across architecture choices on the same dataset, as well as across datasets.

Figure 1 and Figure 3 can also be found in Appendix C.5. As shown, across datasets and image/text tasks, both the performance gain of all DAL strategies over random acquisition, as well as the relative performance among the different DAL acquisition functions, varied substantially. For instance, DAL seemed to demonstrate less benefits over random acquisition on CIFAR10 than the other image datasets, or stronger benefits on AGNEWS than the other text datasets. This is confirmed by the relative average ranking of acquisition function (Appendix C.1.1) across different networks (RESNETs and VGGs for image data and BERT, ROBERTA, DISTILBERT for text data), showing that different acquisition functions have different benefits depending on the datasets. Given the same DNN architecture, FASHION-MNIST (and DBPEDIA

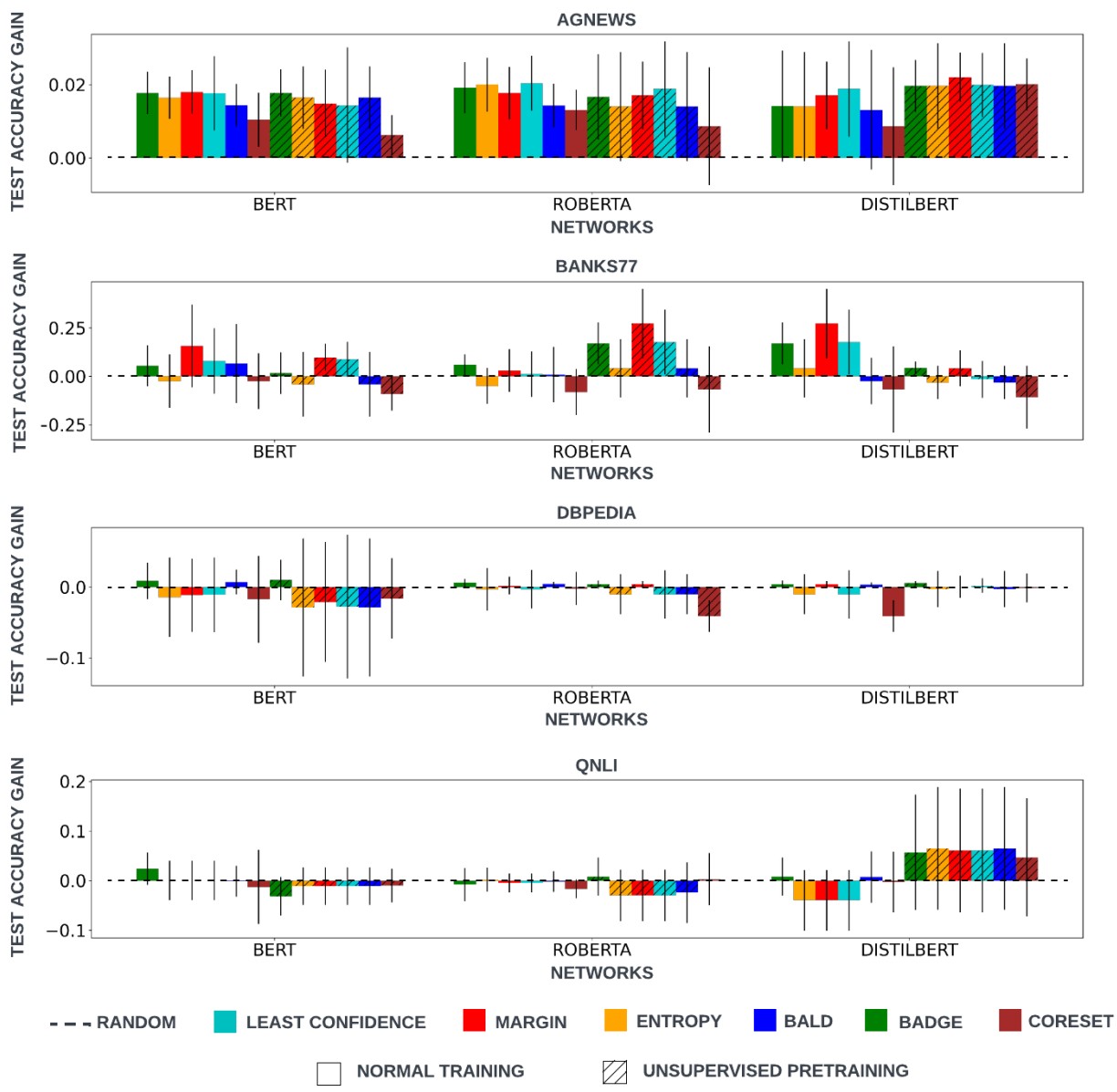

Figure 4: Comparison of gain in test-accuracy over random acquisition (as 0.00 test accuracy gain), for all six acquisition function with (hatched) and without unsupervised pretraining across four datasets and all transformer architectures on text tasks. Similar to Figure 3, results show that the relative performance of acquisition strategies varies both across architecture choices on the same dataset, as well as across datasets.

and QNLI) seemed to induce a more evident performance difference among the acquisition function tested, compared to CIFAR10 and SVHN (AGNEWS and BANKS77) where the performance gap among the tested acquisition functions were small. More importantly, even within the same dataset, the relative performance among the acquisition functions varied with DNN types and sizes: for instance, on MNIST (in Figure 2), data acquisition with least confidence criteria appeared to be the most advantageous using the larger RESNETs (RESNET34 and RESNET50), yet coreset based data acquisition appeared to outperform the others on RESNET18 and the variants of VGGs; similarly, on QNLI (in Figure 4), the hybrid BADGE acquisition seemed to be favored on the BERT architecture followed by DISTILBERT, while becoming the less competitive on the ROBERTA architecture. Furthermore, the use of DISTILBERT in general seemed to induce

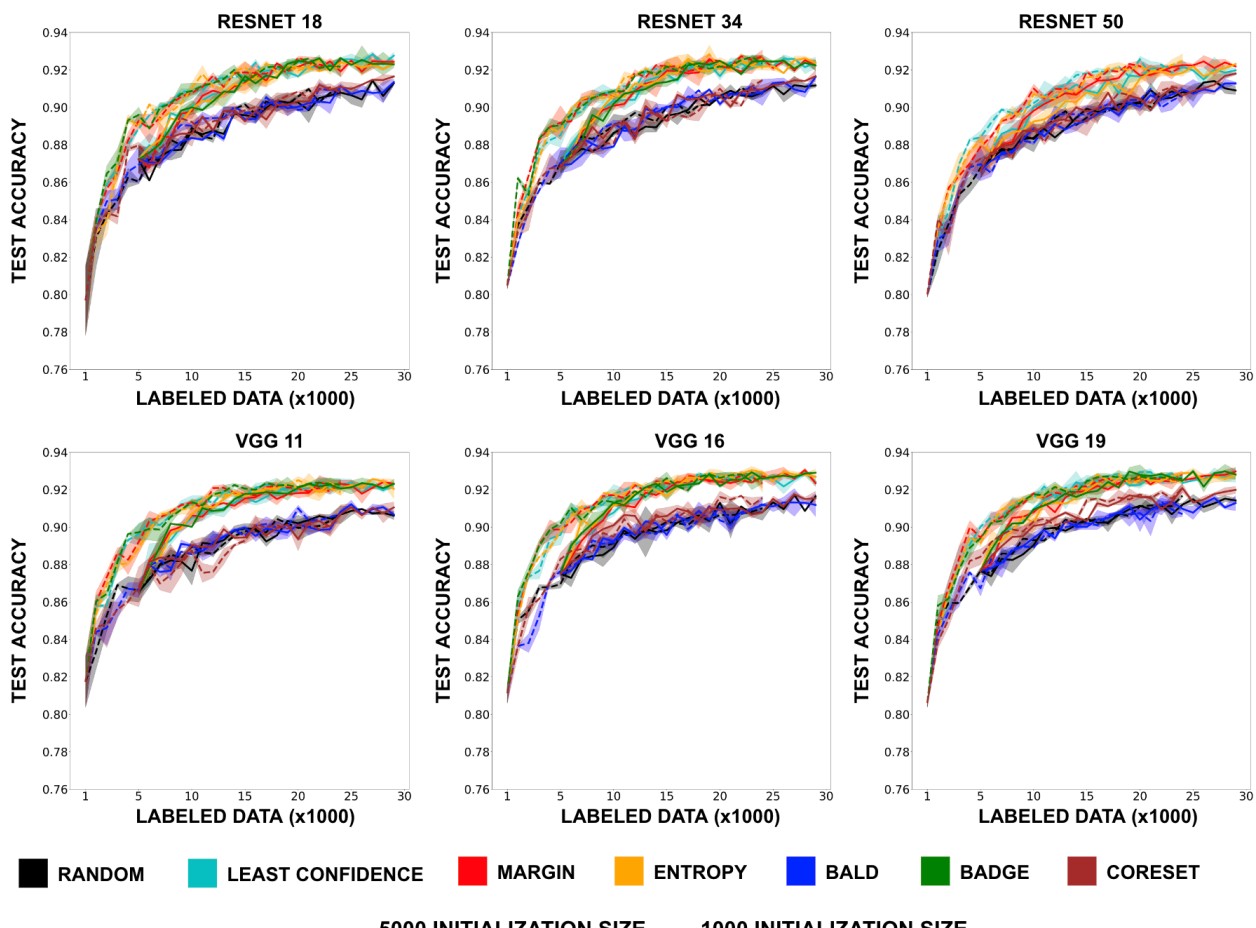

Figure 5: Comparison of test accuracy of different acquisition functions for different DNNs trained on FashionMNIST with an initialization size of 1000 (dashed) *vs.* 5000 (solid). Similar results on other datasets can be found in the Appendix C.3.

larger gaps among the different acquisition functions compared to the other two architectures on the same datasets, although favoring different acquisitions on different datasets (least confidence based acquisistion on AGNEWS, margin based acquisistion on BANKS77, and BADGE on DBPERDIA and QNLI). Note that depending on the metric used (labeling efficiency *vs.* gain in test accuracy), the relative performance among the various acquisition functions as well as their gain over random were also different.

The use of pre-training also appeared to not only influence the relative performance of the tested acquisition functions, but also induced different gains over random acquisition in a architecture-dependent fashion, positively on some architectures and datasets (*e.g.*, RESNET50 on CIFAR10, RESNET18 on FASHION-MNIST, ROBERTA on AGNEWS and BANKS77, and DISTILBERT on QNLI), and negatively on some (*e.g.* all VGGs on MNIST, all RESNETs on SVHN, and BERT on AGNEWS and BANKS77).

Ultimately, no acquisition function consistently outperformed others across datasets and architecture. This indicates that the optimality of data selection strategy depends on the choice of DNN architectures.

**Effect of initialization and acquisition size:** We further varied the initial label size between 1000 and 5000, and the acquisition size among 500, 1000, 1500. As shown in Figure 5, the increase in the initial labeled data size did not show a benefit compared to the smaller initialization size. Increasing the initial labeled data size effectively shifts the performance curve to the right compared to a smaller initial labeled data size. We observe that the performance of BADGE, Entropy, and Margin was worse compared to a lower initialization

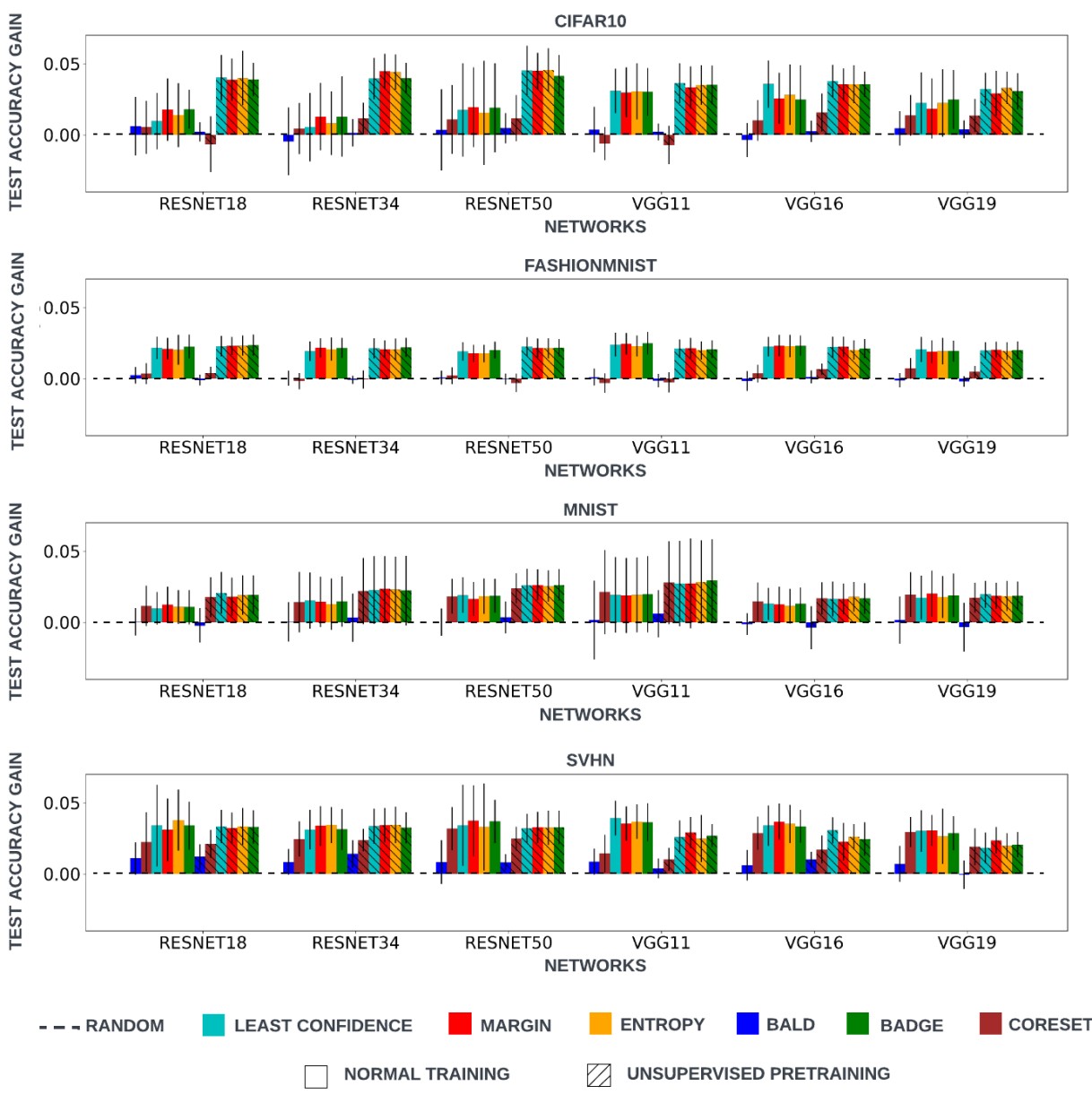

Figure 6: Gain in test-accuracy over random acquisition (as 0.00 test accuracy gain) for different acquisition functions across DNN architectures on four datasets, with (hatched) and without data augmentation. With data augmentation, most acquisition functions exhibited a similar gain over the random acquisition, and their relative performance appeared to be more clearly separated into two groups (high- *vs.* low-performing).

size at the same number of labeled data while that of BALD, and Coreset were unchanged. The change in acquisition size did not produce noticeable differences on the results (see Appendix C.4).

**Effect of data augmentation:** Fig. 6 shows the test accuracy gain over random acquisition achieved by the same six acquisition functions when we applied data augmentation, including horizontal flips and random crops, to all experiments. Compared to the results without augmentation, two main differences can be observed: with data augmentation, 1) most acquisition functions tested exhibited either a substantially larger gain over the random acquisition (*e.g.*, MNIST for RESNETs and VGG11), or a reduced error in the

gain (SVHN for all networks). CIFAR10 saw both effects across all networks. 2) the gap in the performance among different acquisition functions appeared to be minimal (*e.g.*, BADGE, entropy, least confidence, and margin) across all datasets, in comparison to the no augmentation counterpart. For FashionMNIST, the difference between with and without augmentation appears to be minimal. Additionally (not shown on Fig. 6), consistent to existing findings (Beck et al., 2021; Mittal et al., 2019), the performance gain obtained by data augmentation was more substantial than the use of DAL.

## 4 Focus 2: DNN Architecture Optimization during DAL

### 4.1 Methodology

In this Focus, we investigate how DAL may be affected by the optimization of the architecture of DNN feature extractors. We consider three approaches to optimize the DNN architecture, namely supervised joint-training, supervised pre-training, and unsupervised pre-training. In *supervised joint-training*, we utilize the labeled data that are increasingly acquired during DAL. This translates to a setting that is similar to (Geifman & El-Yaniv, 2018), where DNN architecture and weight parameters are simultaneously optimized as DAL proceeds. In *supervised pre-training*, we utilize the initial labelled data to optimize the DNN architecture prior to active learning. In *unsupervised pre-training*, we utilize the unlabeled data to optimize DNN architecture prior to active learning. The pre-trained DNN is then used to initialize DAL during which the DNN's weight parameters are updated while the optimized architecture is kept fixed. This was motivated by the recent DAL work that advocated for unsupervised DNN pre-training (Siméoni et al., 2021) but on fixed architectures. In all three settings, we consider three representative architecture optimization approaches.

**DNN architecture optimization approach during DAL:** We consider three approaches to optimize the DNN architecture during DAL. In *supervised joint-optimization of DNN architectures and data acquisition,* at each acquisition round within DAL, we iterated between data selection given the choice of acquisition function, and the optimization of the CNN architectural and weight parameters given the new data. In *unsupervised pre-optimization (UPO) of DNN architectures,* we adopt the idea from a recent work (Siméoni et al., 2021; Caron et al., 2018) that pre-trains a DNN by iteratively clustering all unlabeled data and using the obtained clusters as pseudo-labels to train the DNN. In *supervised pre-optimization (SPO) of DNN architectures*, we use a classification task on initial labelled data to pre-train the DNN. While the original work (Siméoni et al., 2021; Caron et al., 2018) utilized this to pre-train the weight parameters of the DNN, we use this to simultaneously optimize the architecture and weight parameters of the DNN.

The optimized DNN architecture is then kept fixed while the pre-trained weight parameters are used to initialize the DNN at each DAL acquisition.

**DNN architectures:** The computational cost associated with architecture optimization is high, especially for complex architectures such as RESNET and transformers. Existing works in architecture inference or optimization on transformers are also limited. Therefore we consider CNN as the choice of architecture in Focus 2.

**DNN architecture optimization:** We consider three methods for architecture optimizations, including two Bayesian inference methods and one NAS method, considering mainly the computational feasibility in including these methods in DAL.

For NAS, we follow the PDARTS method described in (Chen et al., 2019) that defines an over-complete network with $L$ cells each with $N$ nodes. Each node signifies a feature layer and two nodes are connected by operations $o \in O$. We define each subsequent node $x_j$ as the linear combination of operations on node $x_i$ defined by architecture parameter $\alpha^{i,j}$.:

$$x_j = \sum_{i<j} \sum_{o \in O_{i,j}} \frac{exp(\alpha_o^{i,j})}{\sum_{o' \in O} exp(\alpha_{o'}^{i,j})} o(x_i) \tag{5}$$

The optimization is split multiple steps where for each step $u$, the number of operations in the operations space is reduced in comparison to previous step $v$. In addition, the number of skip-connects are controlled by setting dropout rate which increases as we go deeper. As described in (Chen et al., 2019), the architecture parameters are optimized on validation data and the weights of the optimal network are tuned on training data. Training time varies with the amount of data available during DAL, ranging approximately from 1.5 hours using 1000 data samples to 8 hours for 30000 data samples.

We also include two Bayesian architecture inference methods described in (KC et al., 2021) (Depth-Dropout) and (Lee et al., 2018) (BBDropout) that translates to the optimization of the number of CNN layers (depth) and the number of CNN filters in each layer (width). In Depth-Dropout, the hidden layer is observed as:

$$h_l = \sigma(W_l \odot h_{l-1}) \bigotimes z_l + h_{l-1} \tag{6}$$

and for BBDropout, we have:

$$h_l = \sigma((W_l \odot h_{l-1}) \bigotimes z_l) + h_{l-1} \tag{7}$$

where $h_l$ is the feature map of the $l^{th}$ hidden layer, $W_l$ is the weight matrix for CNN filters in the $l^{th}$ layer, $z_l$ is the activation mask for $l^{th}$ layer, $\odot$ is a convolution operation, and $\sigma(.)$ is an activation function. We define a beta process as a prior over the number of hidden layers. A beta process sample can be denoted as $(h_l, \pi_l)$, where $\pi_l \in [0,1]$ denotes the activation probability of a hidden layer function $h_l$. A stick-breaking construction of the beta process can be represented as:

$$\pi_l = \prod_{j=1}^{l} v_j, \quad v_l \sim Beta(\alpha, \beta) \tag{8}$$

For BBDropout, in contrast, the Indian Buffet Process (IBP) is defined over neurons per hidden layer *i.e.*,

$$\pi_l = Beta(\alpha_l, \beta_l) \tag{9}$$

We then define a conjugate Bernoulli process inducing layer-wise binary vectors $z_l$ to drop out neurons per layer. The prior over the network structure variable $Z$ can thus be formulated as

$$p(Z, v|\alpha, \beta) = p(Z|v)p(v|\alpha, \beta) = \prod_{m=1}^{M} Bern(z_{ml}|\pi_l) \prod_{l=1}^{\infty} Beta(v_l|\alpha, \beta) \tag{10}$$

The variational distribution with truncation number of layers $K$ to approximate the true posterior distribution for the Depth-Dropout method is defined as:

$$q(Z, v|a_{l=1}^{K}, b_{l=1}^{K}) = q(Z|v)q(v) = \prod_{m=1}^{M} ConBern(z_{ml}|\pi_l) \prod_{l=1}^{K} Beta(v_l|a_l, b_l) \tag{11}$$

where $ConBern$ is a continuous relaxation of Bernoulli distribution. For simplicity, the number of filters in each layer is limited to $M$.

In BBDropout, the posterior is approximated with the variational distribution of form:

$$q(Z, \pi|X) = \prod_{m=1}^{M} q(\pi_k)q(z_m|\pi_m) \tag{12}$$

where we use Kumaraswamy distribution (Kumaraswamy, 1980) for $q(\pi_m)$ and continuous relaxation of Bernoulli distribution for $q(z_m|\pi_m)$ (Lee et al., 2018; Maddison et al., 2016; Jang et al., 2016; Gal et al., 2017a).

The distinction lies in the utilization of a beta process and its corresponding Bernoulli process independently by DepthDropout, enabling the inference of both the number of layers and nodes per layer. On the other hand, BBDropout marginalizes the beta process, inferring the number of nodes in each layer but not the depth. For both DepthDropout and BBDropout, the evidence lower bound (ELBO) of the marginal likelihood of observed data $D$ can be derived and optimized via structured stochastic variational inference (SSVI) as described in (KC et al., 2021).

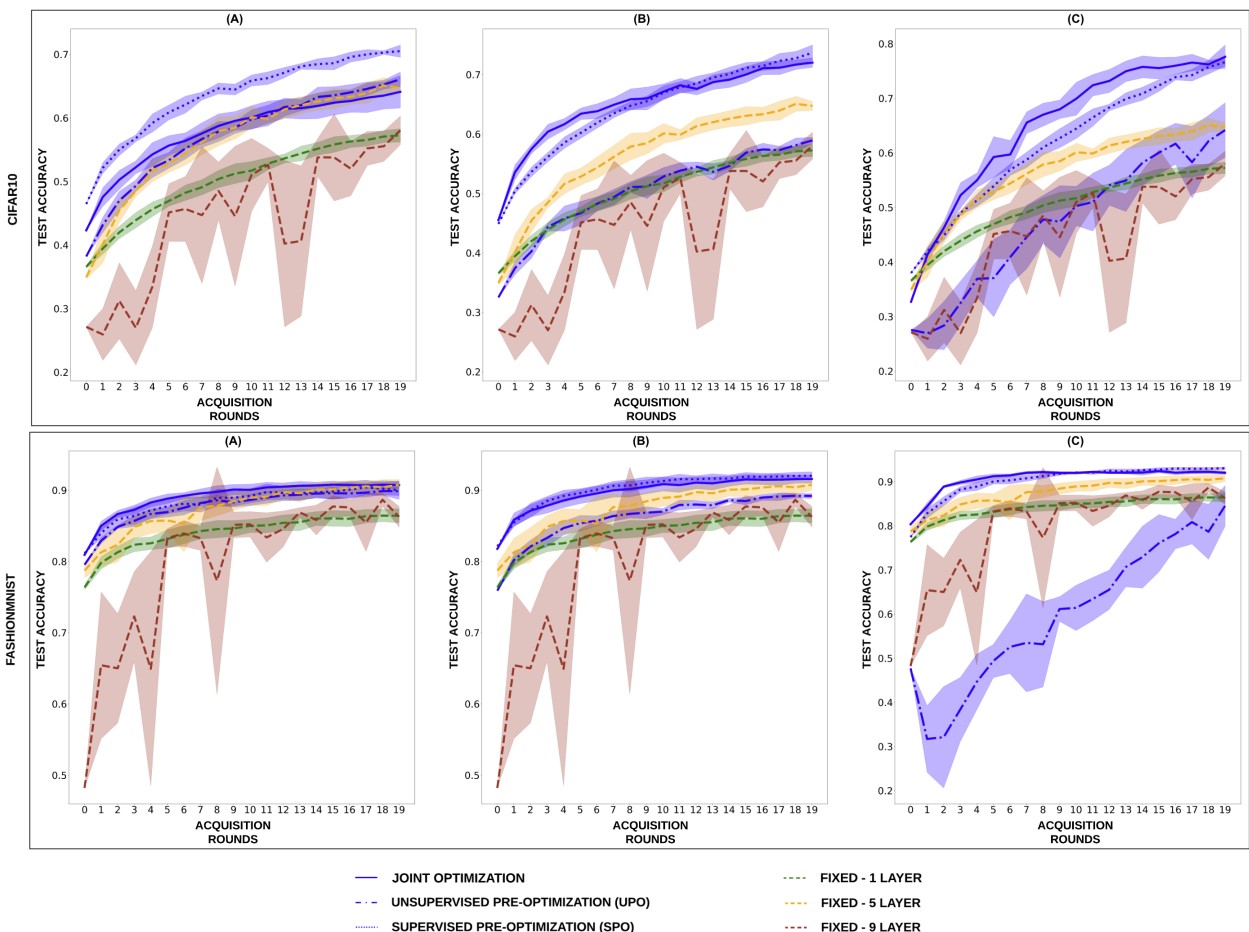

Figure 7: Spread of performance across different acquisition functions (described in Section 3) given pre-defined (fixed) CNN architecture or jointly-optimized, supervised pre-optimized (SPO), unsupervised pre-optimized (UPO) using Depth-Dropout (A), BBDropout(B) and PDARTs(C) for CIFAR10 (top row) and FashionMNIST (bottom row) . Optimization of the DNN architecture, either jointly or during pre-training, in general improved over pre-defined fixed 1, 3 and 5 layer CNN networks performance.

**Evaluation metrics:** Here we focus on the relative performance of DAL on fixed *vs.* optimized DNN architectures, as well as the relative contribution of DNN architecture optimization *vs.* data acquisition optimization to the final DNN performance. We measure this by the *spread of performance* due to the optimization of one factor when the other is controlled. This is visualized as well as quantified as the absolute difference between the maximum and minimum accuracy achieved when one of these two factors varies.

## 4.2 Experiments and Results

Experiments in Focus 2 were performed on four image datasets including MNIST (Deng, 2012), Fashion MNIST (Xiao et al., 2017), CIFAR10 (Krizhevsky, 2009b), and SVHN (Netzer et al., 2011). For Depth-Dropout and BBDropout method, we used CNN with truncation $K = 20$ and 64 filters in each layer. A uniform distribution $U(0, 1.1)$ and $U(0, 1.0)$ was used to initialize the prior of architecture parameters $a$ and $b$ in equation 11 for Depth-Dropout method. The parameters $a$ and $b$ were initialized to 1.3133 and 1.000, respectively, in equation 9 for BBDropout. For PDARTS, we define a total of eight operations (Max pool 3x3, Average Pool 3x3, Skip-connect, Identity, Separable Convolution 3x3 and 5x5, Dilated Convolution 3x3 and 5x5) which are dropped off with dropout rate of 0.1, 0.4 and 0.7 in the subsequent training steps

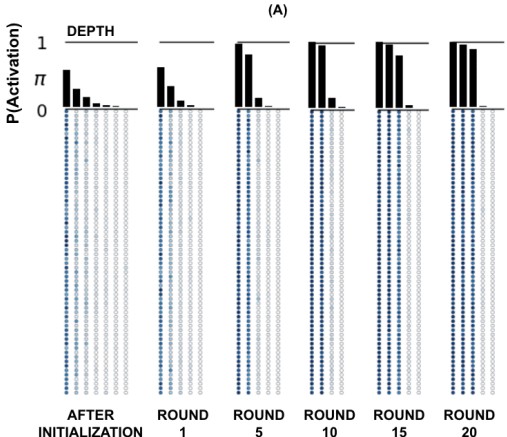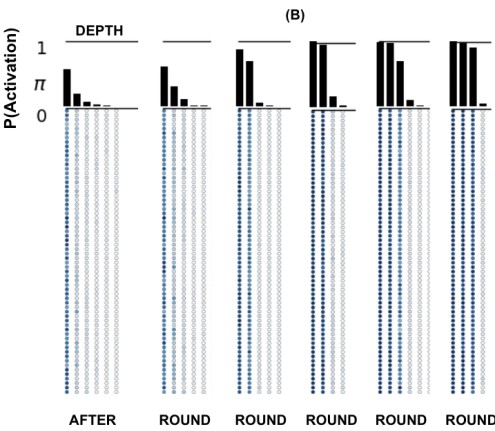

Figure 8: Visualization of optimized CNN architecture as DAL proceeds using Depth-Dropout. A: CIFAR10; B: FashionMNIST.

as used in Chen et al. (2019). All the models were trained for 500 epochs. Experiments were performed on workstations with RTX 2080Ti GPU and 32 GB of RAM as well as P8 and V100 GPU provided by Research Computing at Rochester Institute of Technology (Rochester Institute of Technology, 2024). The active learning experiments are run on four image dataset in same setting as described in 3.2.

**The effect of optimized DNN architectures:** Fig. 7 summarizes the DAL performance across different acquisition functions when applied to architecture optimization – Depth-Dropout (A), BBDropout (B), and PDARTs(C) – and fixed DNN architectures with 1, 5, and 9 convolutional layers (between input and output layers of the network) on CIFAR10 and FashionMNIST. The fixed architectures are used as baselines to compare with the networks with optimized architectures. The DNN architectures were optimized either jointly or during pre-training using both supervised and unsupervised methods. Complete results on the rest of the datasets can be found in Appendix D.1. The figure depict that the optimized networks, particularly those optimized jointly or through supervised pre-optimization (SPO) approach, consistently outperform fixed pre-defined networks. SPO and joint optimization approaches, in general, showed consistent gains in all architecture optimization methods (i.e. Fig. 7A, B and C) in comparison to unsupervised pre-optimization (UPO) approach which performed least favorably. This may be attributed to the fact that the network optimization with unlabelled data was based on a clustering task different for primary DAL task.

In case of joint optimization approach, the continuous change in the size of labeled data during DAL resulted in changes in optimal DNN architecture. This, in turn, induces continuous modifications to the optimal architecture, complicating the joint optimization with data selection. To corroborate this, we experimented with more extreme changes of data size by considering an initial size of 200 and acquisition size of 100 in the same experiments. Figure 8 (A) and (B) shows the optimized architecture using Depth Dropout as DAL proceeds for CIFAR10 and FashionMNIST respectively, where the bar graph on top indicates the probability of activation of a layer and the column below indicates the activation of filters in each layer. As shown, both the depth and width of the CNN increased as DAL proceeded with adding labeled data. These results suggested that the joint optimization approach via small and growing labeled data in addition to weight optimization and active learning may complicate optimal data acquisition due to the continued change in the architecture, whereas leveraging initial labeled data to pre-optimize the feature space may offer a simpler and yet more competitive solution.

The limited performance with unsupervised pre-training using PDARTS may be attributed to the sensitivity of the unsupervised labeling task to the number of clusters used for generating pseudo-labels. Through our experiments, we observed that employing a larger cluster size with PDARTS resulted in an optimized network with minimal test accuracy ($< 30\%$), which improved as the number of clusters approached the actual number of labels in the dataset. In contrast, supervised pre-training of PDARTS with an initially

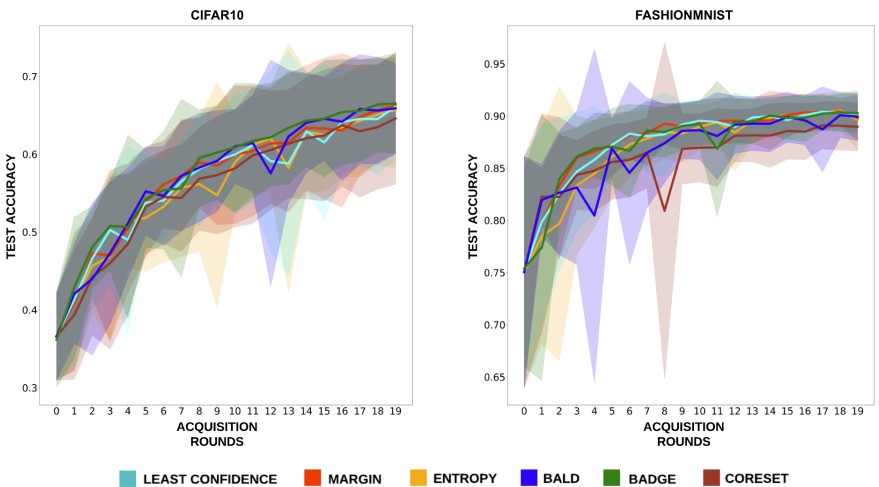

Figure 9: Spread of performance across different architectures – pre-defined architecture and architeure optimization of CNN architectures – given acquisition functions for CIFAR10 and FashionMNIST, in the form of spread of test accuracy over the course of DAL. Comparison with Fig. 7 shows that the optimization of DNN architecture outweighed the effect of data selection in the overall DAL performance. Similar results for MNIST and SVHN are shown in Appendix D.2

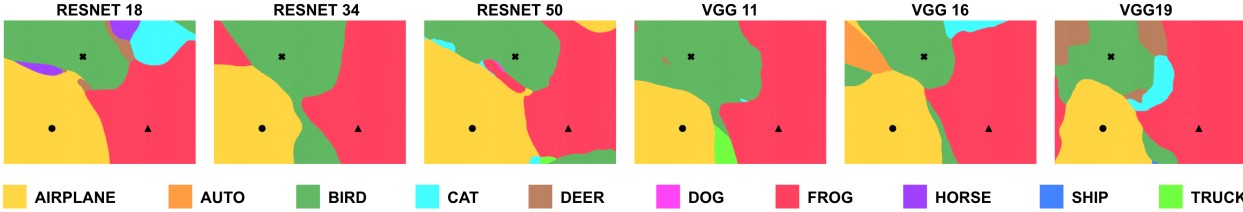

Figure 10: The class boundaries of six architectures (RESNET 18/34/50 and VGG 11/16/19) shown across columns, plotted on the plane spanning three randomly selected images (● - Airplane, × - Bird, ▲ - Frog classes) in CIFAR10 dataset for a single seed. The figure shows that change in architecture introduces change in the decision boundary.

labeled dataset yielded a significantly superior optimized network compared to the unsupervised approach. This suggests that unsupervised pre-optimization of DNN architectures may be difficult.

**Relative contribution of architecture *versus* data optimization:** The shade or spread in Fig. 7 describe the spread of performance between different acquisition functions for different architecture optimizations. A closer look into Fig. 7 suggest that, with the exception of UPO on PDARTS, the performance spread across different acquisition functions is reduced using an optimized DNN architecture compared to fixed pre-defined networks. Figure 9 further summarizes the spread of performance across all optimized or fixed DNN architectures, for each given acquisition function on CIFAR10 and FashionMNIST. Complete results on the rest of the datasets can be found in Appendix D.2. As shown, different choices of architecture parameters induced a large performance gap of the DNN at any given data size for any acquisition function used. This performance spread changed as DAL proceeded, although the trend of change was not consistent among datasets: on FashionMNIST, the gap among different CNN architectures appeared to be larger at the earlier stages of DAL when the data size was smaller, whereas on CIFAR this gap appeared to increase as DAL proceeded Contrasting Fig. 7 with Figure 9, it is evident that the impact of the DNN architectures substantially outweighed and even reduced the impact of acquisition strategies. This further suggests that, when the labeling budget is small, the effort to identify optimal architecture of DNN feature extractor may be critical in order to maximize the efficacy of active data selection.

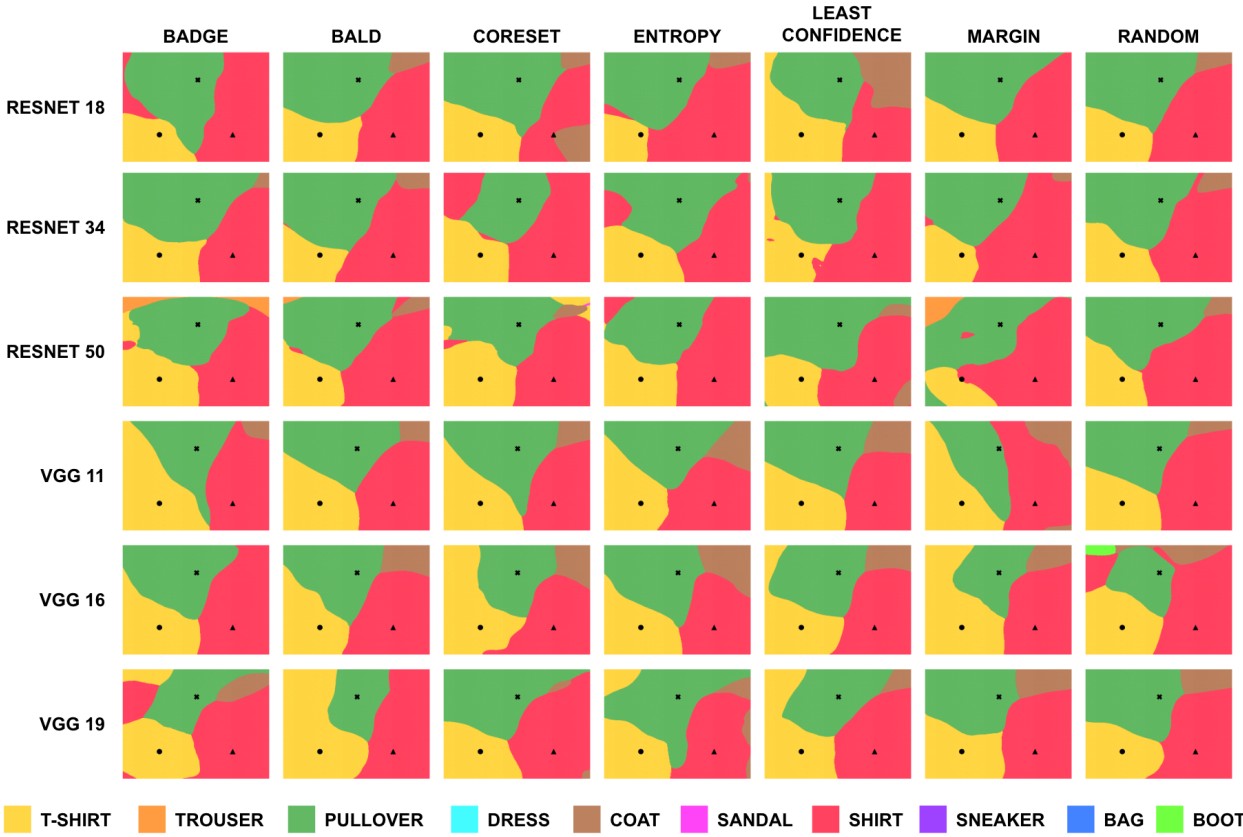

Figure 11: The class boundaries of six architectures (RESNET 18/34/50 and VGG 11/16/19) along the rows trained with seven acquisition functions shown across columns, plotted on the plane spanning three randomly selected images (● - T-Shirt, × - Pullover, ▲ - Shirt) in FashionMNIST for a single seed. The decision boundary for the three example classes vary both across the networks as well as acquisition functions. Complete result for remaining image dataset are added in appendix.

## 5  Discussion

### 5.1  Relation between DAL acquisition, DNN architecture, and decision boundaries

What may explain the observed interdependence between the DNN architecture and data acquisition? (Kolossov et al., 2023) showed in their work that a better performing model in general is not always an ideal choice during active learning. Recent studies (Mickisch et al., 2020; Lei et al., 2023) showed that the task decision boundary of a network changes continuously during training to generalize to the available data. (Lei et al., 2023) showed that the variability in the decision boundary of network inversely affects the generalization and reproducibility of the results. Furthermore, when we consider different network architectures to train on the same data, the decision boundary appears to visibly vary (Somepalli et al., 2022). Illustration of this is shown in Figure 10 where the class decision boundary of six architectures is plotted on the plane spanning three randomly selected images (Plane, Frog and Bird) of CIFAR-10. We further extended the visualization in Figure 10 to include different acquisition functions shown for FashionMNIST in Figure 11. Complete results on remaining image datasets can be found in Appendix C.2. The decision boundary appeared to change with both acquisition function as well as the underlying network architecture considered. The change in decision boundary appeared more prominent with change in the network architecture in comparison to the change in acquisition function.

To further understand the effect of change in decision boundary on data acquisition, we added a simple experiment on half moon dataset with MLP (Multi Layer Perceptron) architectures of varying sizes (1,2

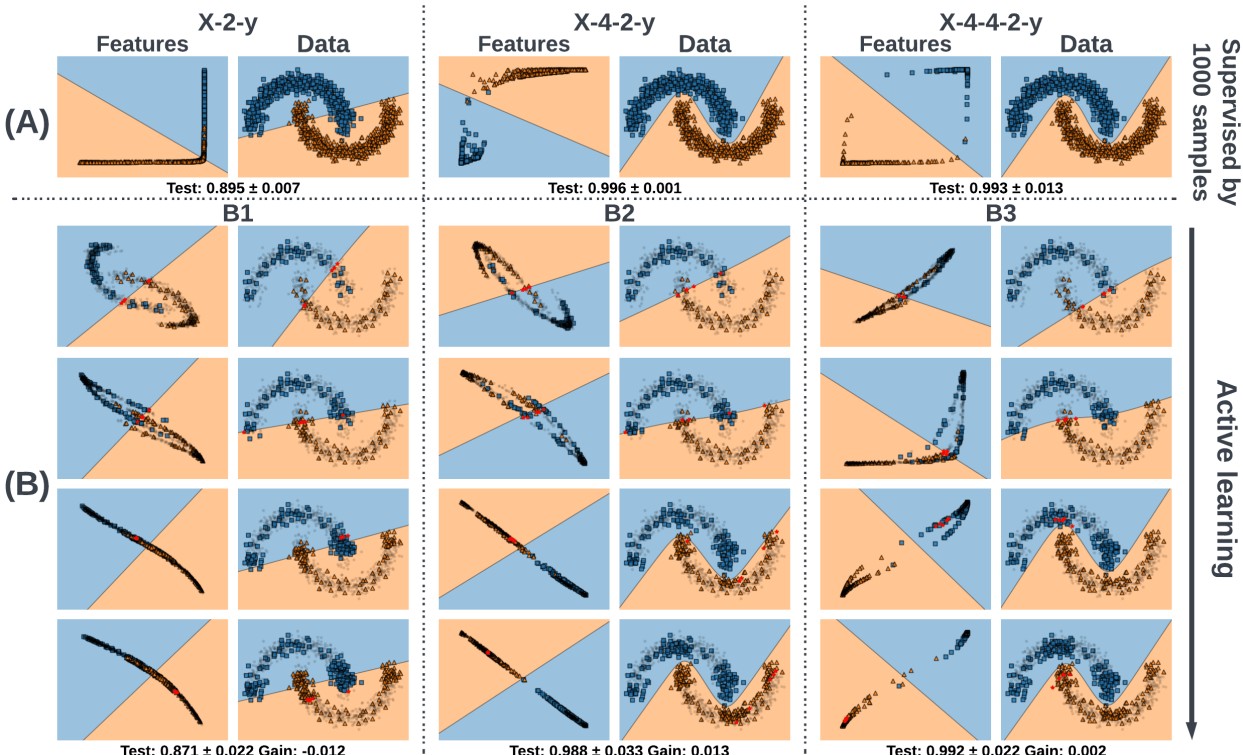

Figure 12: Illustrations of the change of optimal decision boundary due to differences in the DNN feature space (A) and its impact on DAL data selection and gain over random acquisition of data (B). Each column (separated by dotted vertical lines) represents decision boundaries achieved by different architecture of neural networks visualized in the feature (left) and data (right) space, where 2, 4-2, 4-4-2 in between input x and output y denote the number of nodes used in each hidden layer. The figures show that change in architecture changes the features and thus optimal decision boundary, which in turn affects optimal data selection in DAL.

and 3 layers) in Figure 12. We observed, as the architecture of the model changed, the *optimal* decision boundary changed accordingly. When aiming to *actively* capture such decision boundary with a small amount of carefully-selected labeled data (using Entropy), the differences in feature space as shown in Figure 12B had a clear impact. First and foremost, data selection for the same acquisition strategies (red) became very different, even for overly expressive DNNs (Fig. 12B2-3) that eventually arrived at similar decision boundaries. Furthermore, the gains of active learning over random acquisitions not only became substantially different among the various DNN architectures, but also appeared marginal compared to the performance gap induced by architecture differences.

Since optimal decision boundary plays an important role in data acquisition (Kim et al., 2021; Tharwat & Schenck, 2023) , it may be the mechanism through which DNN architectures affects data acquisition. Future works investigating the theoretical underpinning for this relation may shed light on how to best address the interdependence between these two optimizations in DAL.

## 5.2 Effect of the dataset on data acquisitions

Additionally, though not explicitly mentioned, different dataset differ from each other owing to the complexity of the dataset which includes underlying structure of dataset, dimensionality, noise, redundancy in data, decision boundary complexity, etc. These affect several factors that are important to the design of acquisition functions, including the uncertainty of a DNN that will affect uncertainty-based acquisition functions, the diversity of the data samples that will affect diversity-based acquisition functions, and the decision boundary which will affect all acquisition functions. Recent work (Kim et al., 2021) has shown that the ranking of

acquisition functions vary depending on the nature of the dataset (balanced / imbalanced). Similarly, the scalability of acquisition functions has also been found to change with dataset (Ji et al., 2023). Additionally, many existing works Mittal et al. (2019); Mayer & Timofte (2020); Beck et al. (2021); Zhang et al. (2024) in the active learning community have implicitly shown in their experiment that the performance of different acquisition functions differ for different datasets. Our results presented in Figure 1 to Figure 4 and Appendix C.1.1 confirmed this effect. Because the optimal DNN architecture depends on the underlying dataset, this may also contribute to the observed dependence between the DAL performance and choices of DNN architecture.

### 5.3 Limitations and Future work

Investigations in Focus 2 of the current study is focused on relatively small CNN based architecture optimization. To further generalize the findings, future studies need to extend to larger overparameterized architectures such as RESNET, VGG, and transformers. Such extension will provide a more comprehensive understanding of how optimization of different network architectures influence data optimization, providing insights into the applicability of joint architecture and data optimization strategies across a variety of DNN types. Additionally, current work is focused on relatively smaller datasets like MNIST, FashionM-NIST, SVHN and CIFAR10. Broadening the scope of datasets to include larger datasets like CIFAR100 (Krizhevsky, 2009a), Imagenet (Deng et al., 2009), CelebA (Liu et al., 2015), etc will help generalizing the observations across more complex datasets space.

There is also room for incorporating additional data acquisition strategies in the presented study. The current study examined a range of acquisition functions, covering prevalent strategies representative of uncertainty-based, diversity-based, and hybrid strategies found in the existing literature of DAL. Future work can broaden the spectrum of acquisition strategies to more recent strategies, such as those incorporating neural tangent kernels to assess DNN training dynamics (Wang et al., 2022), and those incorporating the concept of semi-supervised learning in DAL.

In terms of approaches for optimizing DNN architectures, we considered a cell-based (NAS) method, specifically PDARTs, along with two Bayesian DNN architecture inference methods. For CNN networks considered in the Focus 2 of this study, the connections between operations within a search cell exhibit a large influence on the architecture's performance. However, cell-based NAS methods are not directly transferrable to transformers that operate on attention mechanism rather than convolution. Future work will incorporate NAS methods for transformers, such as AdaBERT and NAS-BERTs, to further test the generalizability of the findings obtained in Focus 2 regarding the effect of DNN architecture optimization on DAL. Future work can also include additional NAS methods for CNN-based architecture, such as dynamic-exploration DARTs where the dynamic architecture varies the kernel size or network path of CNN according to input data.

Finally, to investigate potential strategies for simultaneously optimizing DNN architecture and data selection, we considered unsupervised pre-optimizaiton of DNN artechitecture prior to DAL, versus supervised joint-optimization of DNN during DAL. Both approaches, however, essentially considered DNN architecture and data optimization as two separate optimization problems with their respective objective functions. An interesting yet much more challenging future research direction may be the developments of joint DNN architecture and data optimization theories, methods, and algorithms that integrate these two optimization objectives in a more coherent formulation.

## 6 Conclusion

In this work, we examine the influence of DNN architectures on optimal data selection in DAL. We show that the choices of DNN architecture substantially influence and outweigh data optimization in DAL, and that its optimization helps increase the benefits of active data selection, with supervised pre-optimization being most beneficial followed by joint optimization. We hope that the findings help inform the research community in improving the reproducibility of DAL evaluations by taking into account the important role of DNN architecture choices in DAL, and in opening up new research avenues that better integrate DNN

architecture optimization to maximize the benefits of active data selection when the labeling budget is limited.

## Acknowledgements

This work is supported by the National Science Foundation funding NSF OAC-2212548 and the NSF award no. 2045804.

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

## A   Appendix

## B   Architectures of RESNET and VGG Networks

### B.1   RESNET

Structure of VGG-11/16/19 and RESNET-18/34/50 are shown below in format Conv x, y where x is filter size and y represents number of filters.

|  | RESNET-18 | RESNET-34 | RESNET50 |
|---|---|---|---|
| CONV LAYER | Conv 3, 64 | Conv 3, 64 | Conv 3, 64 |
| BLOCK 1 | [Conv 3, 64 Conv 3, 64] x 2 | [Conv 3, 64 Conv 3, 64] x 3 | [Conv 1, 64 Conv 3, 64 Conv 1, 256] x 3 |
| BLOCK 2 | [Conv 3, 128 Conv 3, 128] x 2 | [Conv 3, 128 Conv 3, 128] x 4 | [Conv 1, 128 Conv 3, 128 Conv 1, 512] x 4 |
| BLOCK 3 | [Conv 3, 256 Conv 3, 256] x 2 | [Conv 3, 256 Conv 3, 256] x 6 | [Conv 1, 256 Conv 3, 256 Conv 1, 1024] x 6 |
| BLOCK 4 | [Conv 3, 512 Conv 3, 512] x 2 | [Conv 3, 512 Conv 3, 512] x 3 | [Conv 1, 512 Conv 3, 512 Conv 1, 2048] x 3 |
| FULLY CONNECTED LAYERS | FC width 512 FC width C | FC width 512 FC width C | FC width 2048 FC width C |

### B.2   VGG

|  | VGG-11 | VGG-16 | VGG-19 |
|---|---|---|---|
| CONVOLUTION LAYERS | Conv 3, 64 Max-pool | 2 x [Conv 3, 64] Max-pool | 2 x [Conv 3, 64] Max-pool |
|  | Conv 3, 128 Max-pool | 2 x [Conv 3, 128] Max-pool | 2 x [Conv 3, 128] Max-pool |
|  | 2 x [Conv 3, 256] Max-pool | 3 x [Conv 3, 256] Max-pool | 4 x [Conv 3, 256] Max-pool |
|  | 2 x [Conv 3, 512] Max-pool | 3 x [Conv 3, 512] Max-pool | 4 x [Conv 3, 512] Max-pool |
|  | 2 x [Conv 3, 512] Max-pool | 3 x [Conv 3, 512] Max-pool | 4 x [Conv 3, 512] Max-pool |
| FULLY CONNECTED LAYERS | 2 x [FC width 4096] FC width 1000 FC width C | 2 x [FC width 4096] FC width 1000 FC width C | 2 x [FC width 4096] FC width 1000 FC width C |

# C   Focus 1: The Effect of DNN Architectures on DAL Performances

## C.1   Effect of Acquisition Functions

### C.1.1   Ranking of Acquisition function

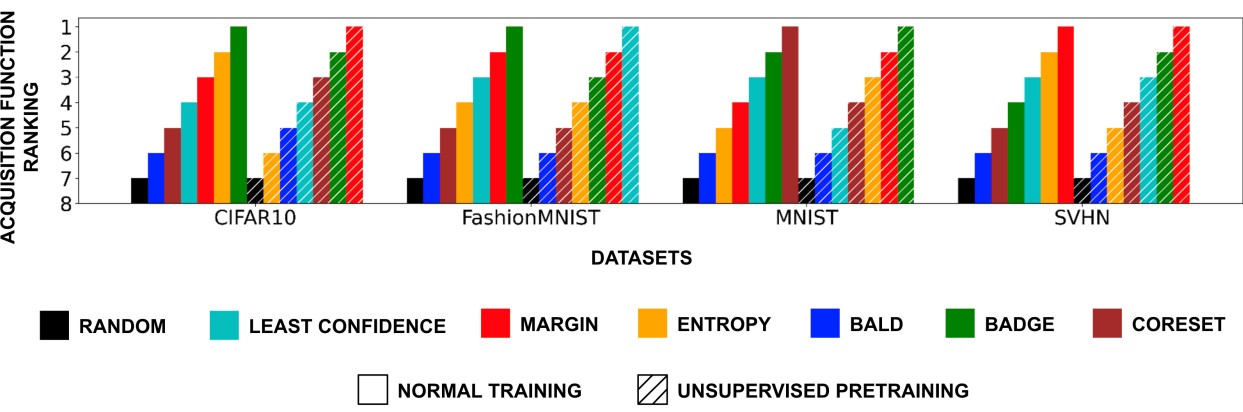

Figure 13: Average ranking plot for acquisition functions for image datasets averaged across six networks (RESNET 18/34/50 and VGG 11/16/19). The ranking of different acquisition functions vary with dataset considered

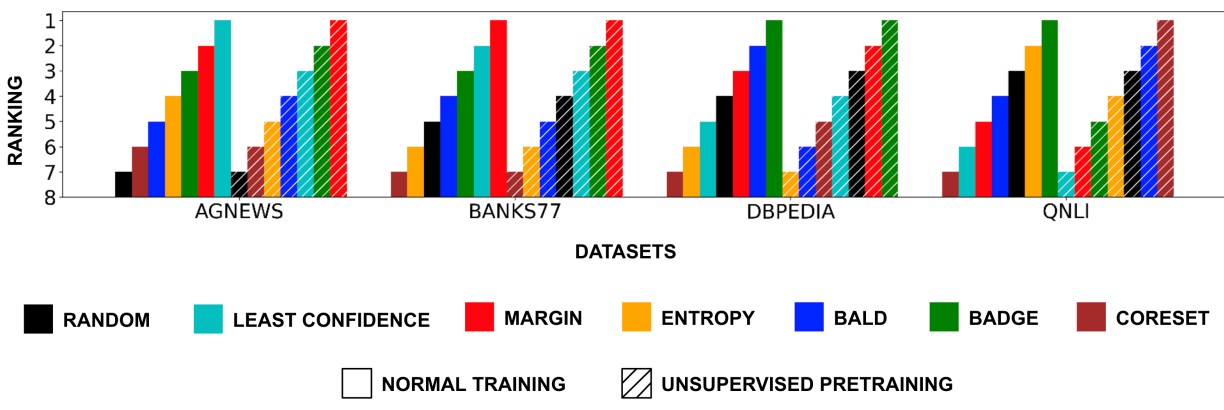

Figure 14: Average ranking plot for acquisition functions for text datasets averaged across three networks (BERT, ROBERTA, DISTILBERT). The ranking of different acquisition functions vary with dataset considered

## C.2 Effect on Decision Boundary

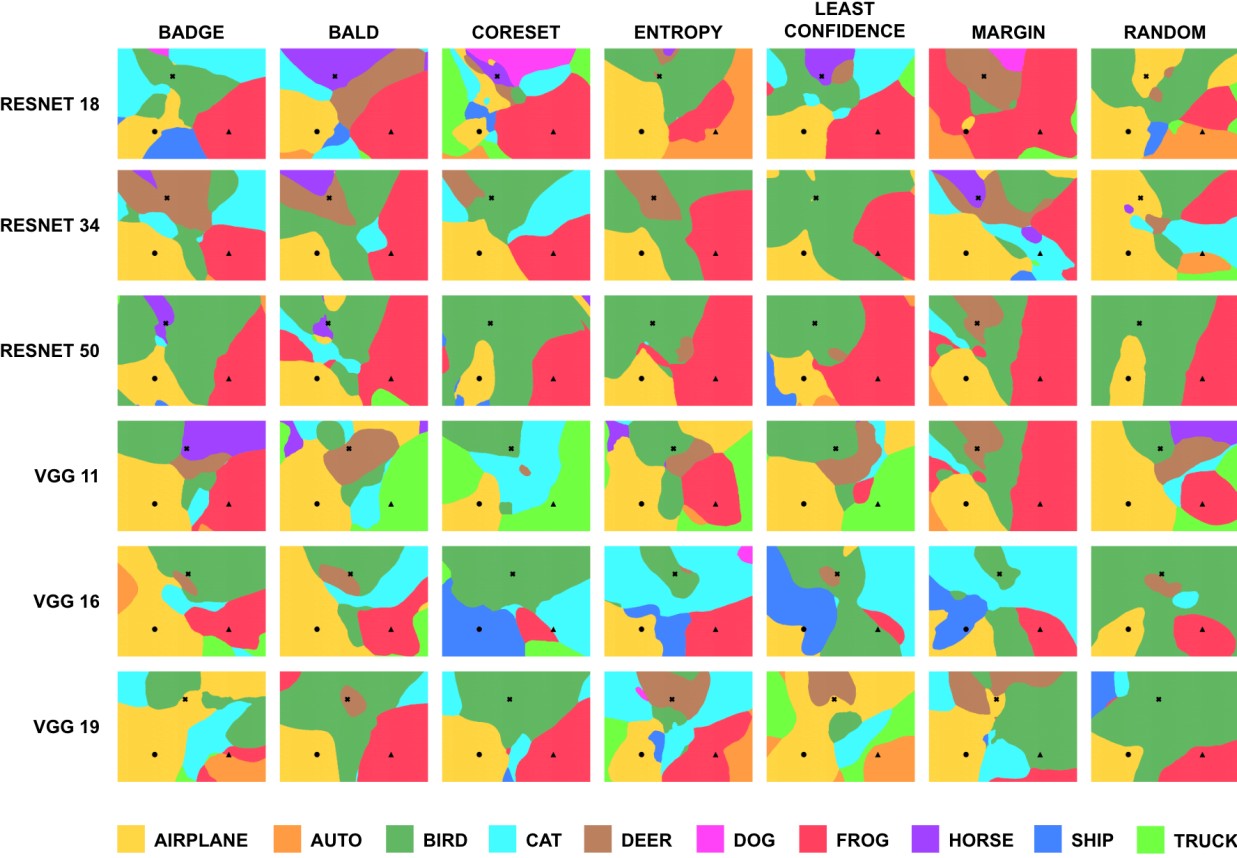

Figure 15: The class boundaries of six architectures (RESNET 18/34/50 and VGG 11/16/19) along the rows trained with seven acquisition functions shown across columns, plotted on the plane spanning three randomly selected images (● - Airplane, × - Bird, ▲ - Frog) in CIFAR10 for a single seed.

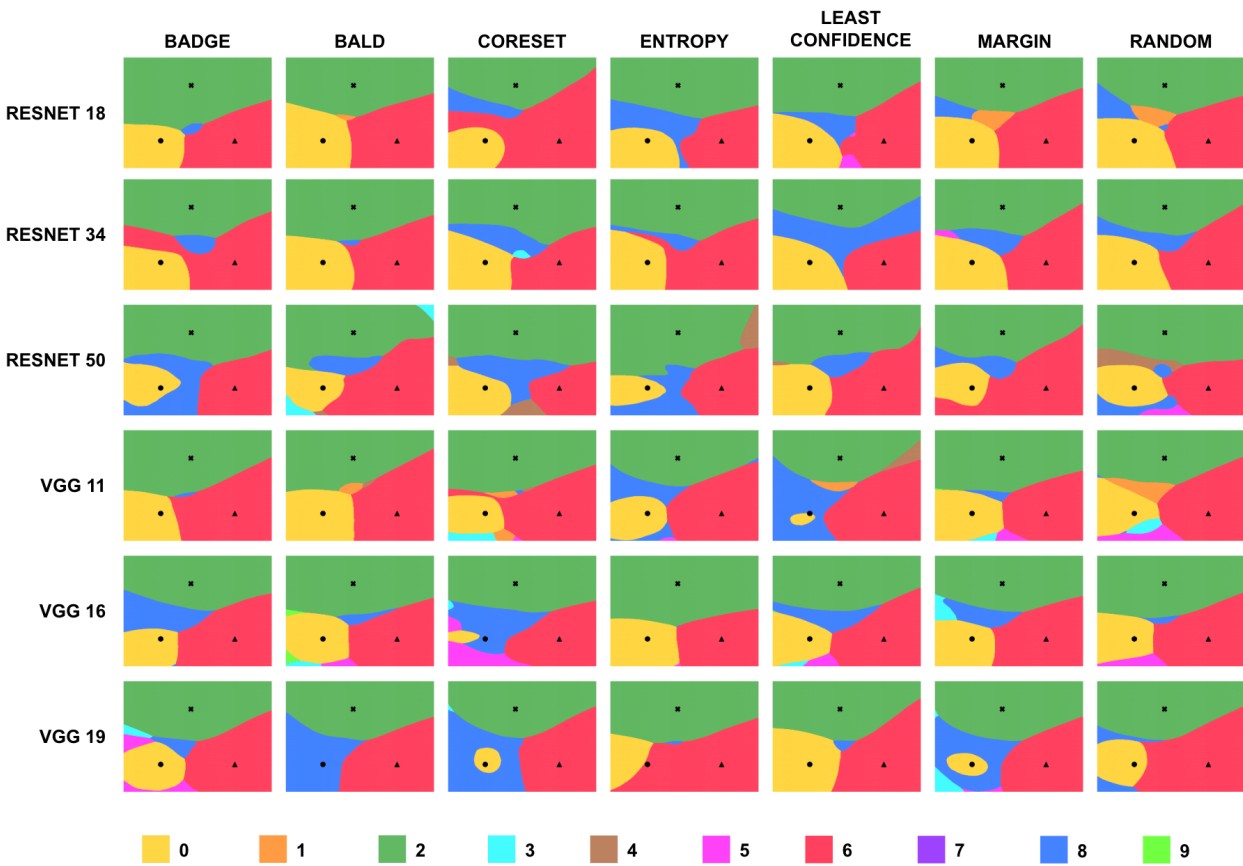

Figure 16: The class boundaries of six architectures (RESNET 18/34/50 and VGG 11/16/19) along the rows trained with seven acquisition functions shown across columns, plotted on the plane spanning three randomly selected images (classes ● - 0, × - 2, ▲ - 6) in MNIST for a single seed.

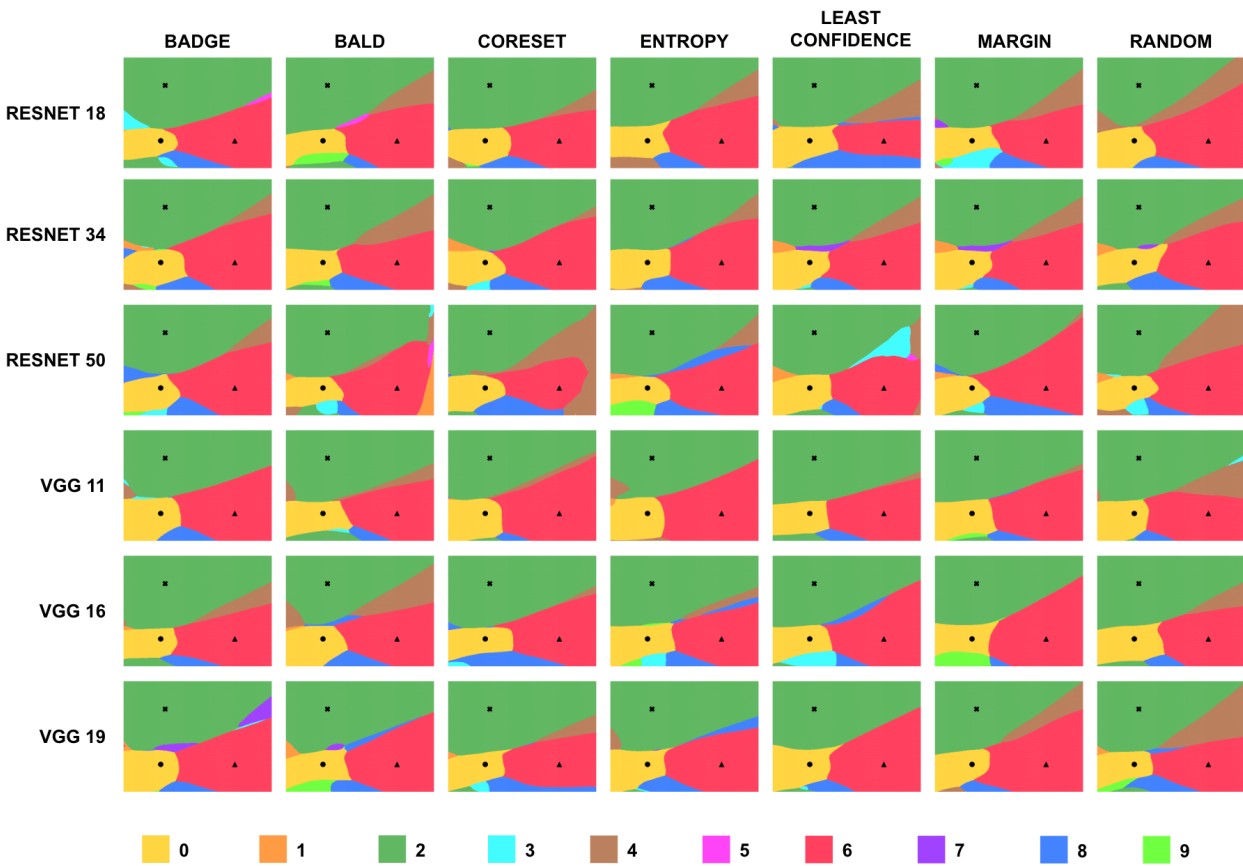

Figure 17: The class boundaries of six architectures (RESNET 18/34/50 and VGG 11/16/19) along the rows trained with seven acquisition functions shown across columns, plotted on the plane spanning three randomly selected images (classes ● - 0, × - 2, ▲ - 6) in SVHN for a single seed.

## C.3    Effect of Initial Training Size

### C.3.1    CIFAR10

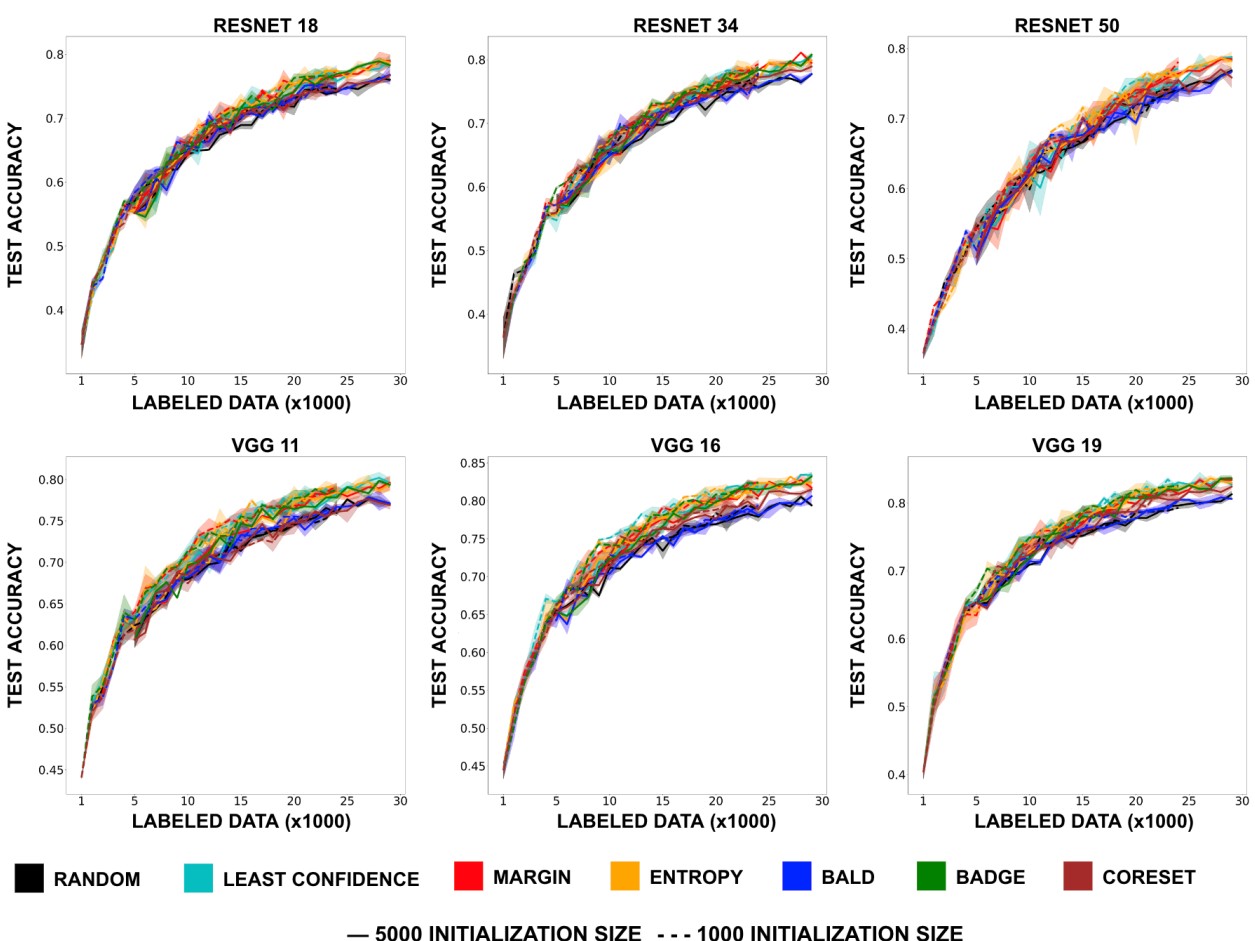

Figure 18: Comparison of test accuracy of different acquisition functions for different network trained on CIFAR10 with initialization size of 1000 (dashed) and 5000(solid).

### C.3.2 FashionMNIST

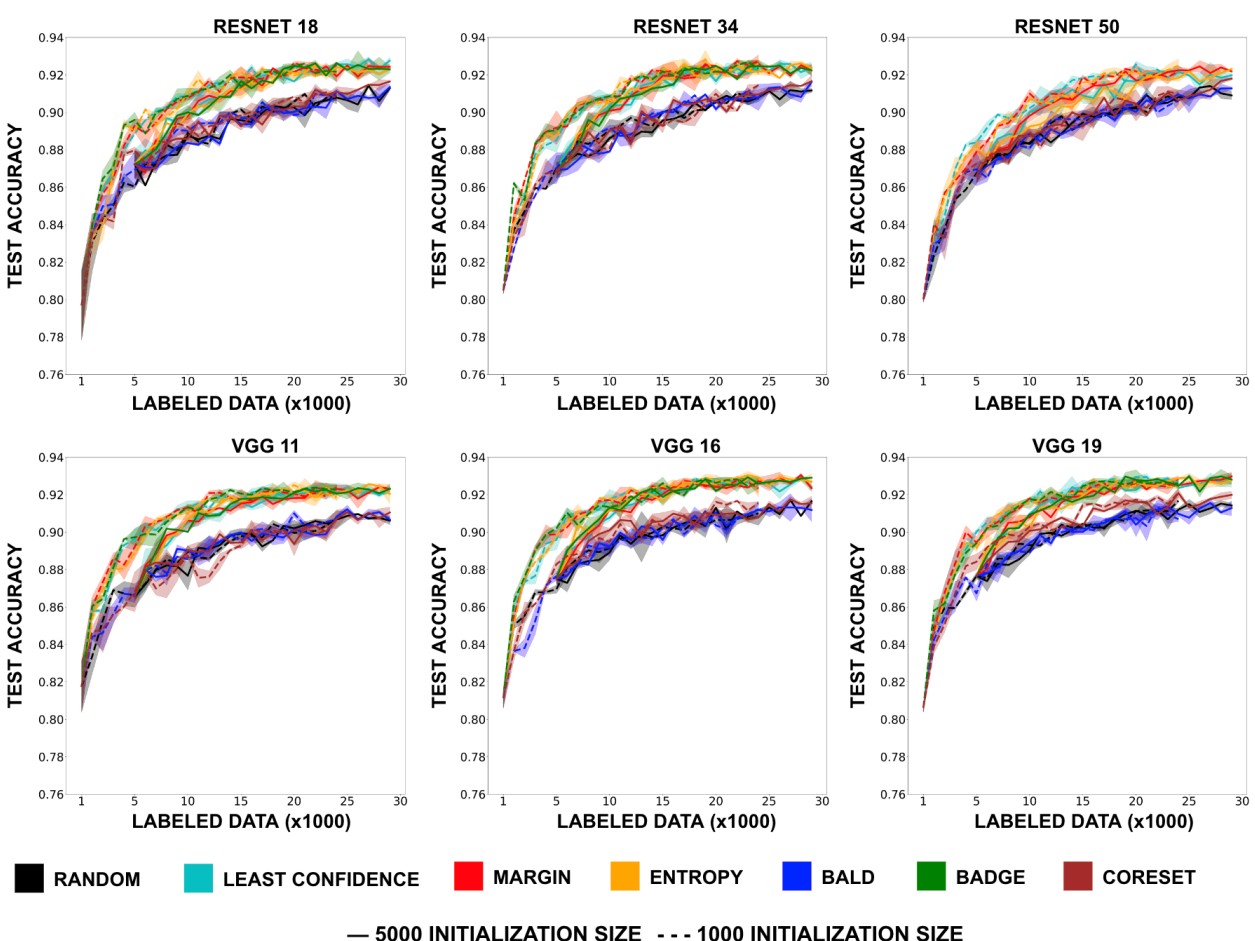

Figure 19: Comparison of test accuracy of different acquisition functions for different network trained on FashionMNIST with initialization size of 1000 (dashed) and 5000(solid).

### C.3.3    MNIST

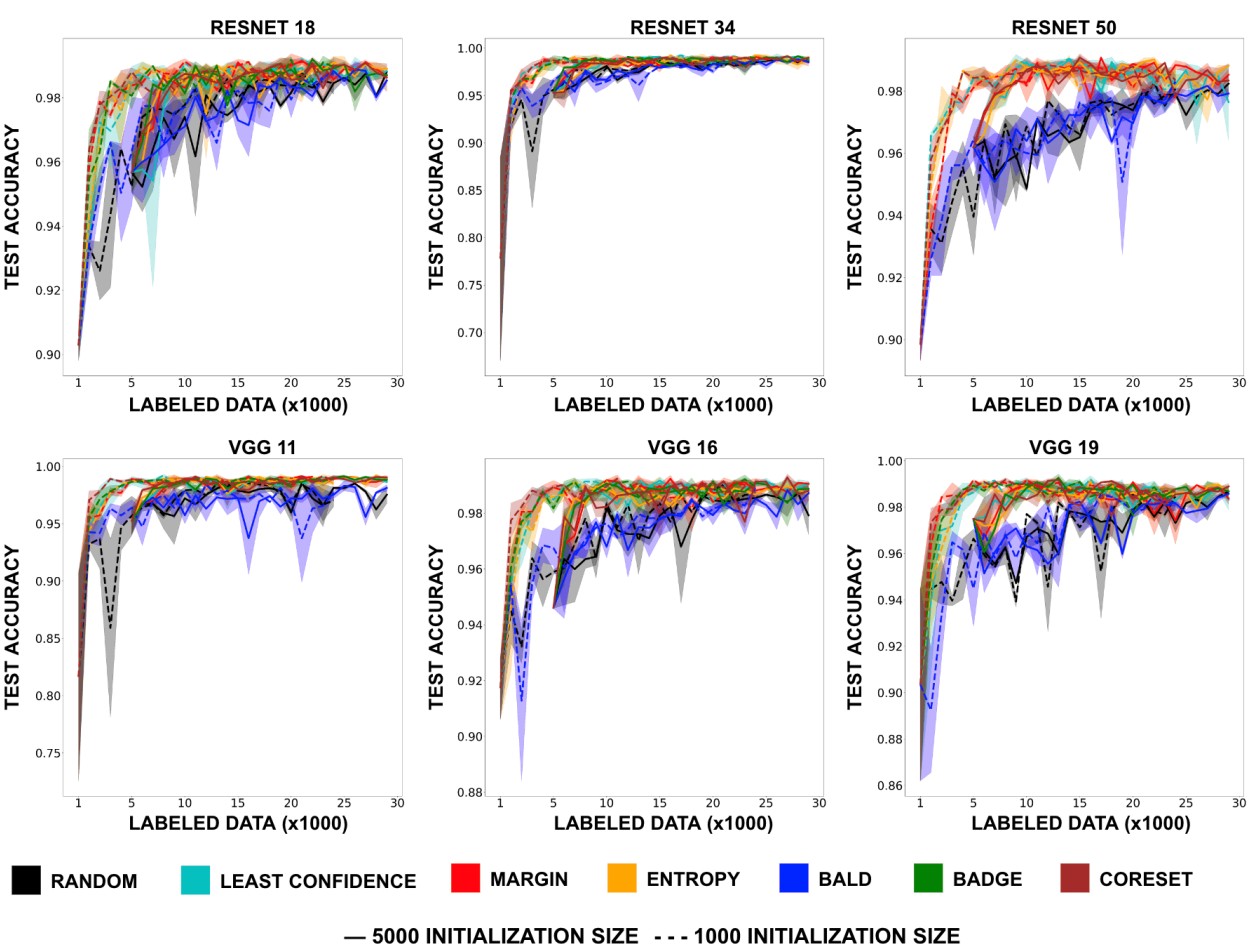

Figure 20: Comparison of test accuracy of different acquisition functions for different network trained on MNIST with initialization size of 1000 (dashed) and 5000(solid).

### C.3.4 SVHN

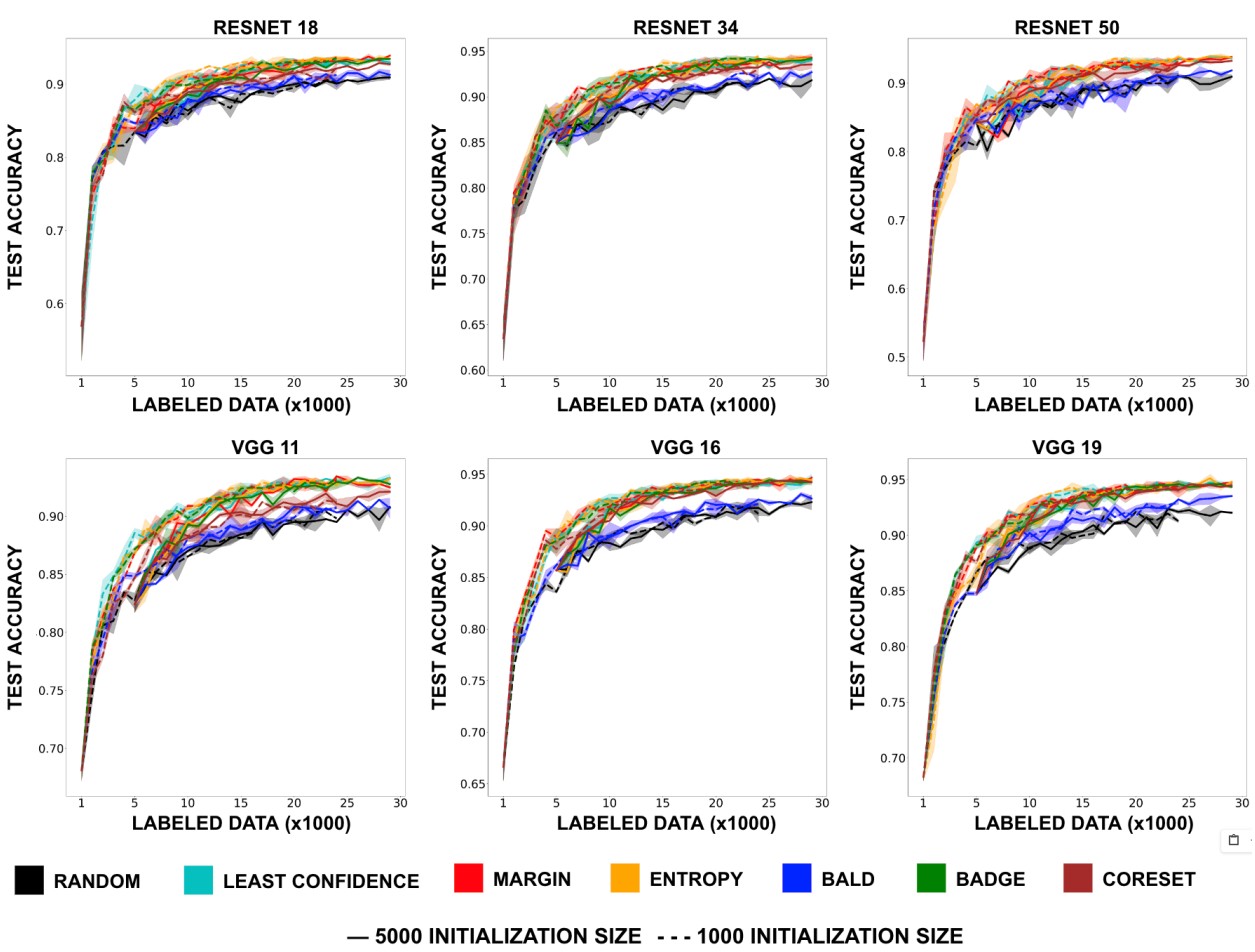

Figure 21: Comparison of test accuracy of different acquisition functions for different network trained on SVHN with initialization size of 1000 (dashed) and 5000(solid).

## C.4    Effect of Acquisition size

### C.4.1    CIFAR10

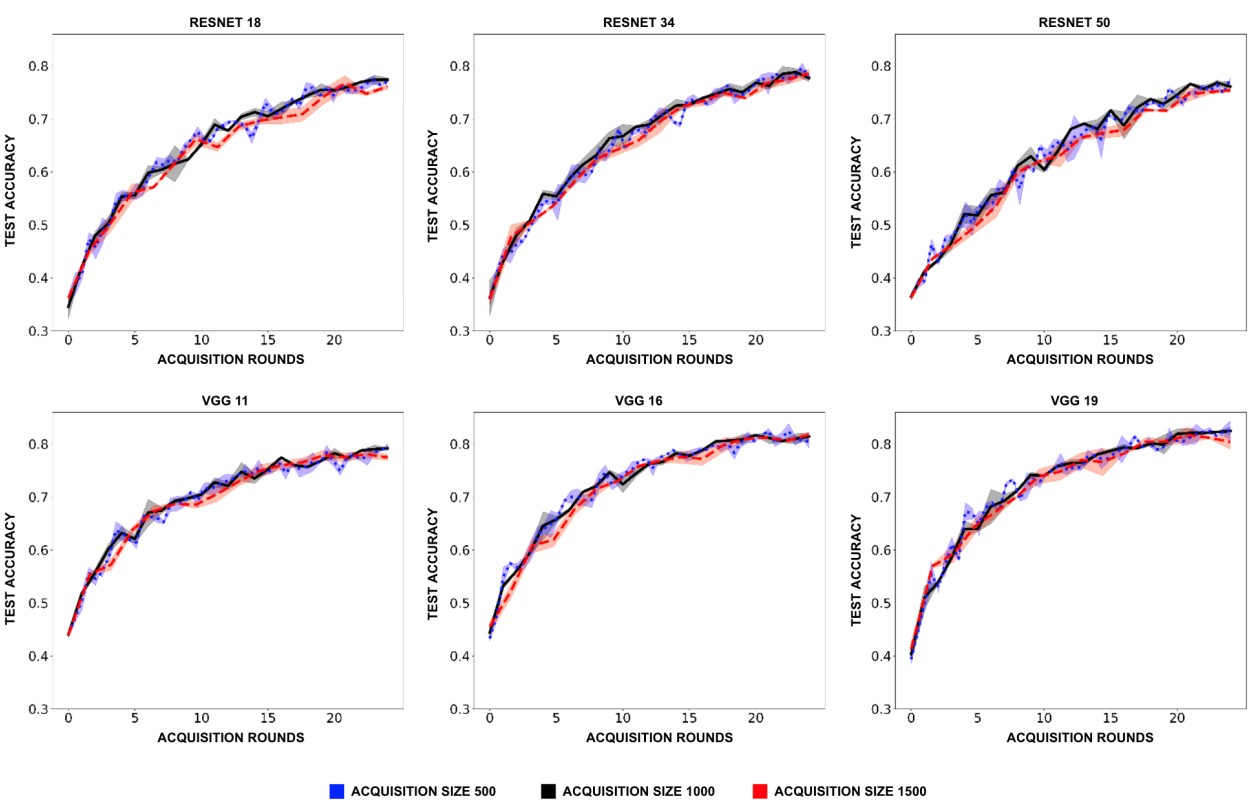

Figure 22: Visualization of performance of Entropy acquisition function on different networks trained on CIFAR10 with acquisition size of 500 (dotted blue), 1000 (solid black) and 1500 (dashed red) respectively

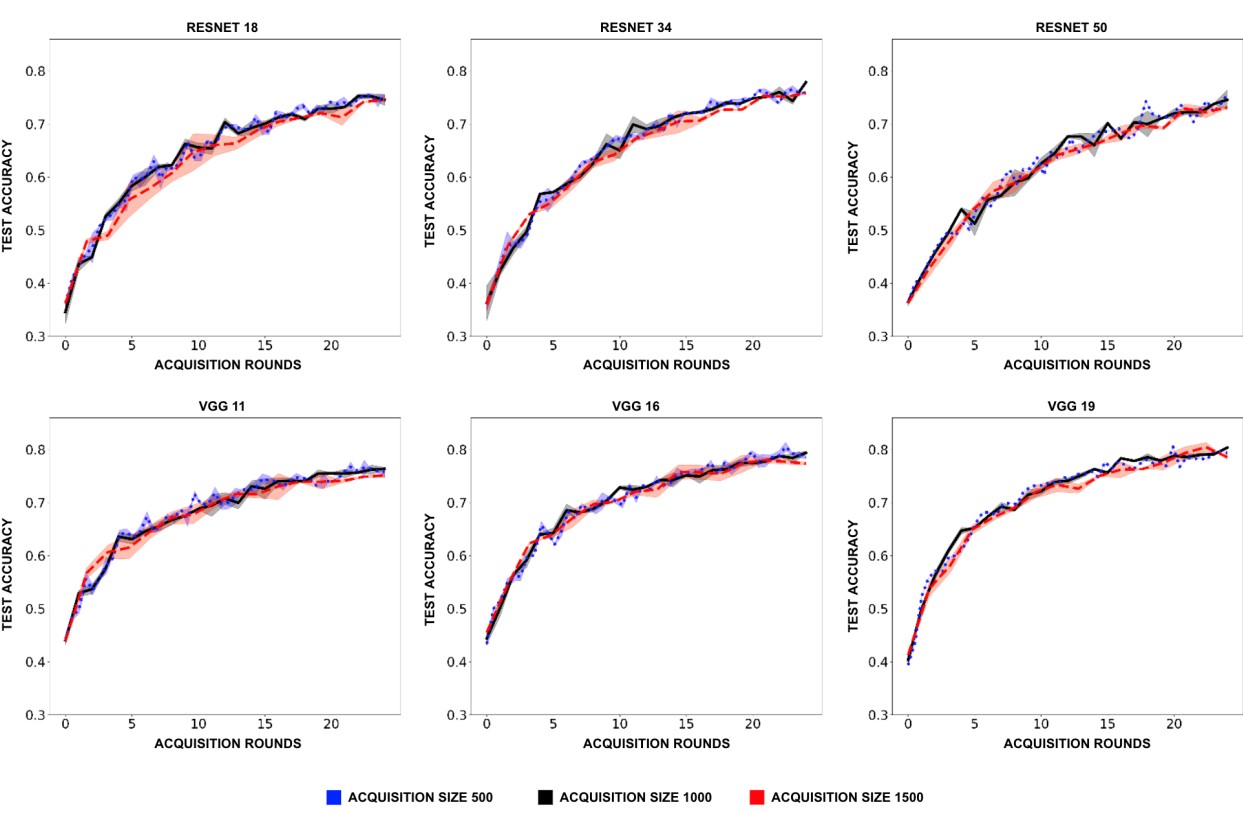

Figure 23: Visualization of performance of BALD acquisition function on different networks trained on CIFAR10 with acquisition size of 500 (dotted blue), 1000 (solid black) and 1500 (dashed red) respectively

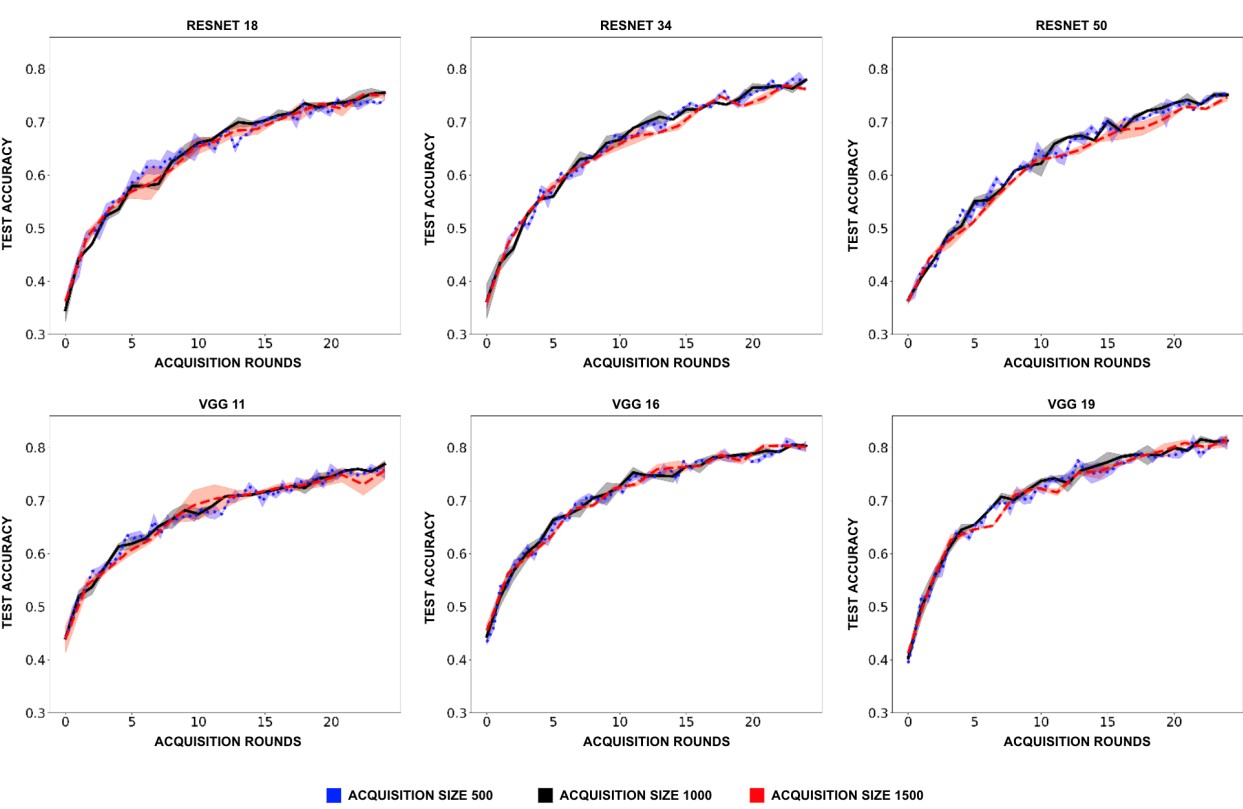

Figure 24: Visualization of performance of Coreset acquisition function on different networks trained on CIFAR10 with acquisition size of 500 (dotted blue), 1000 (solid black) and 1500 (dashed red) respectively

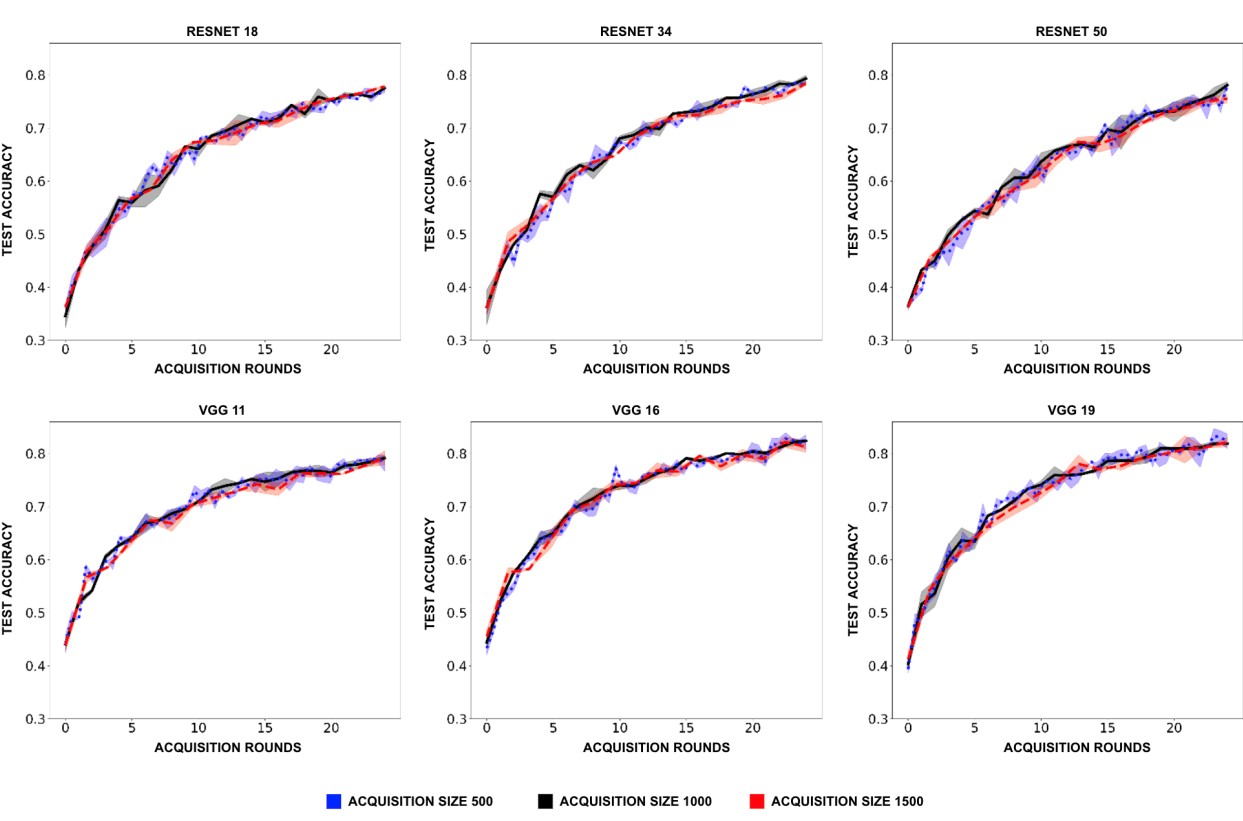

Figure 25: Visualization of performance of Margin acquisition function on different networks trained on CIFAR10 with acquisition size of 500 (dotted blue), 1000 (solid black) and 1500 (dashed red) respectively

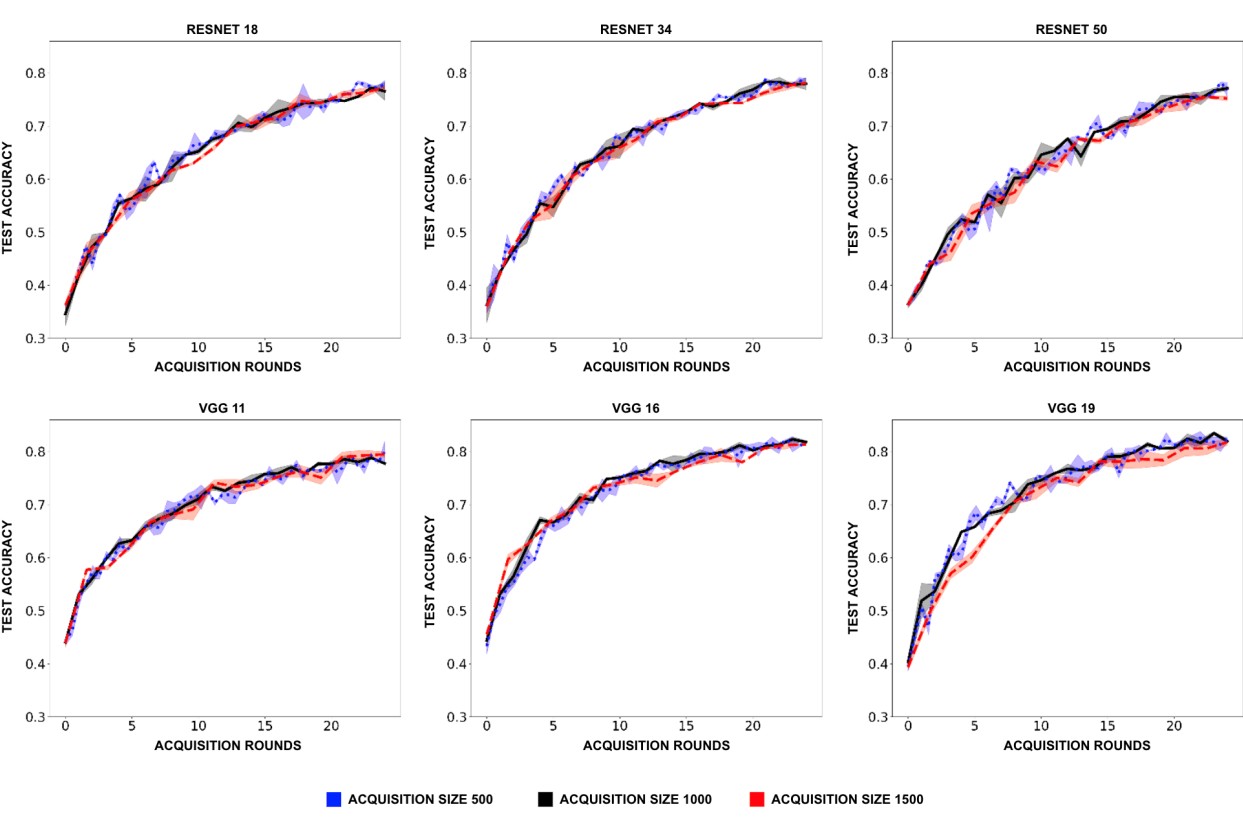

Figure 26: Visualization of performance of Least Confidence acquisition function on different networks trained on CIFAR10 with acquisition size of 500 (dotted blue), 1000 (solid black) and 1500 (dashed red) respectively

### C.5 Tabular representation for Labeling Efficiency

### C.5.1 Image Dataset

**(A)**

| | BALD | Coreset | Least Confidence | Margin | Entropy | BADGE | BALD Pre | Coreset Pre | Least Confidence Pre | Margin Pre | Entropy Pre | BADGE Pre |
|---|---|---|---|---|---|---|---|---|---|---|---|---|
| RESNET18 | 1.0129+/-0.0790 | 1.0205+/-0.0571 | 1.0114+/-0.0681 | 1.0366+/-0.1049 | 1.0338+/-0.0863 | 1.0541+/-0.0607 | 1.0544+/-0.0476 | 0.9892+/-0.0375 | 0.9704+/-0.0904 | 1.0603+/-0.1127 | 0.9693+/-0.1061 | 1.0171+/-0.1046 |
| RESNET34 | 0.9773+/-0.0779 | 1.0227+/-0.0818 | 0.9895+/-0.0899 | 1.0467+/-0.1102 | 1.0093+/-0.0867 | 1.0316+/-0.0980 | 1.0249+/-0.0820 | 1.0006+/-0.0731 | 0.9769+/-0.1247 | 1.0075+/-0.1282 | 0.9565+/-0.1344 | 1.0181+/-0.0861 |
| RESNET50 | 0.9993+/-0.0472 | 1.0263+/-0.0843 | 1.0417+/-0.1039 | 1.0625+/-0.0611 | 1.0118+/-0.1455 | 1.0635+/-0.0916 | 1.0871+/-0.1125 | 1.1088+/-0.1019 | 1.0905+/-0.1629 | 1.1226+/-0.1549 | 1.0849+/-0.1311 | 1.1302+/-0.1527 |
| VGG11 | 1.0093+/-0.0949 | 0.9572+/-0.0347 | 1.1024+/-0.0961 | 1.1073+/-0.1367 | 1.1080+/-0.1266 | 1.1082+/-0.1165 | 1.0450+/-0.0380 | 1.0148+/-0.0629 | 1.0387+/-0.1158 | 1.0912+/-0.0877 | 1.0738+/-0.1112 | 1.0568+/-0.0924 |
| VGG16 | 0.9735+/-0.0328 | 1.0107+/-0.0464 | 1.1105+/-0.1273 | 1.0650+/-0.0792 | 1.0615+/-0.0799 | 1.0428+/-0.1172 | 0.9832+/-0.0337 | 1.0409+/-0.0500 | 0.9973+/-0.0592 | 0.9801+/-0.0711 | 0.9499+/-0.1072 | 1.0107+/-0.1046 |
| VGG19 | 1.0107+/-0.0360 | 1.0335+/-0.0605 | 1.0301+/-0.0948 | 1.0214+/-0.0981 | 1.0280+/-0.1113 | 1.0498+/-0.1132 | 0.9875+/-0.0689 | 0.9922+/-0.0793 | 0.9527+/-0.1067 | 0.9384+/-0.0863 | 0.8918+/-0.0667 | 0.9429+/-0.1153 |

**(B)**

| | BALD | Coreset | Least Confidence | Margin | Entropy | BADGE | BALD Pre | Coreset Pre | Least Confidence Pre | Margin Pre | Entropy Pre | BADGE Pre |
|---|---|---|---|---|---|---|---|---|---|---|---|---|
| RESNET18 | 1.0979+/-0.1347 | 1.0609+/-0.1306 | 1.5662+/-0.3962 | 1.5312+/-0.3481 | 1.4338+/-0.3751 | 1.5865+/-0.3756 | 1.3084+/-0.2172 | 1.3302+/-0.1520 | 1.7586+/-0.3666 | 1.7655+/-0.3527 | 1.7061+/-0.4594 | 1.6378+/-0.3774 |
| RESNET34 | 1.0094+/-0.0887 | 0.9878+/-0.1281 | 1.4471+/-0.3552 | 1.5405+/-0.3918 | 1.4420+/-0.3954 | 1.5240+/-0.4709 | 1.1848+/-0.1320 | 1.4167+/-0.3336 | 1.5447+/-0.3491 | 1.5010+/-0.2478 | 1.4617+/-0.3888 | 1.5185+/-0.2535 |
| RESNET50 | 1.0255+/-0.0651 | 1.0638+/-0.1362 | 1.4714+/-0.3654 | 1.3898+/-0.2342 | 1.3955+/-0.2496 | 1.5400+/-0.2972 | 1.2996+/-0.2416 | 1.2990+/-0.2820 | 1.5669+/-0.4420 | 1.6065+/-0.4771 | 1.4930+/-0.3334 | 1.5255+/-0.4192 |
| VGG11 | 1.0468+/-0.0878 | 0.9851+/-0.1323 | 1.7027+/-0.5668 | 1.7682+/-0.5532 | 1.6768+/-0.5147 | 1.7649+/-0.5591 | 1.0984+/-0.1823 | 1.4049+/-0.3437 | 1.5984+/-0.4452 | 1.5718+/-0.4853 | 1.5980+/-0.4929 | 1.6952+/-0.5340 |
| VGG16 | 0.8988+/-0.1678 | 1.0058+/-0.1344 | 1.5441+/-0.6030 | 1.5727+/-0.5920 | 1.4851+/-0.5108 | 1.6132+/-0.5694 | 1.1267+/-0.1778 | 1.4308+/-0.4016 | 1.4917+/-0.4614 | 1.4859+/-0.5695 | 1.4239+/-0.4080 | 1.4863+/-0.4858 |
| VGG19 | 1.0004+/-0.0908 | 1.1367+/-0.2570 | 1.4564+/-0.4955 | 1.4350+/-0.4477 | 1.3947+/-0.4752 | 1.4229+/-0.3895 | 1.0243+/-0.0799 | 1.5327+/-0.4309 | 1.5755+/-0.4837 | 1.5657+/-0.4630 | 1.4657+/-0.4540 | 1.5083+/-0.4140 |

**(C)**

| | BALD | Coreset | Least Confidence | Margin | Entropy | BADGE | BALD Pre | Coreset Pre | Least Confidence Pre | Margin Pre | Entropy Pre | BADGE Pre |
|---|---|---|---|---|---|---|---|---|---|---|---|---|
| RESNET18 | 1.0644+/-0.3094 | 1.8425+/-0.3874 | 1.3908+/-0.1626 | 1.6801+/-0.3848 | 1.5664+/-0.2399 | 1.5123+/-0.4870 | 1.2516+/-0.2242 | 1.3404+/-0.2205 | 1.4382+/-0.3994 | 1.7171+/-0.6495 | 1.9429+/-0.9723 | 1.6563+/-0.5685 |
| RESNET34 | 0.9554+/-0.2157 | 1.4396+/-0.5525 | 1.6727+/-0.5949 | 1.4484+/-0.3886 | 1.4754+/-0.4511 | 1.5600+/-0.4362 | 1.1802+/-0.2108 | 1.7202+/-0.5914 | 1.4898+/-0.4466 | 1.4721+/-0.4372 | 1.5941+/-0.5667 | 1.7205+/-0.5095 |
| RESNET50 | 0.9051+/-0.2577 | 2.1179+/-0.9303 | 2.4425+/-1.0581 | 1.8728+/-0.5519 | 2.1188+/-0.8667 | 2.2862+/-0.7590 | 1.2392+/-0.3157 | 1.4583+/-0.3460 | 1.4211+/-0.4773 | 1.5277+/-0.4172 | 1.3939+/-0.3390 | 1.5112+/-0.4802 |
| VGG11 | 0.6231+/-0.3388 | 2.2346+/-1.0206 | 2.0251+/-0.9600 | 1.8571+/-0.7553 | 1.9359+/-0.9462 | 2.1174+/-1.1010 | 1.1705+/-0.2458 | 1.4903+/-0.3722 | 1.5207+/-0.4335 | 1.3378+/-0.3174 | 1.5682+/-0.5321 | 1.3770+/-0.8344 |
| VGG16 | 1.2136+/-0.3291 | 2.2716+/-1.4983 | 1.7956+/-0.8938 | 1.9915+/-1.1283 | 1.7128+/-0.9412 | 1.9897+/-1.0545 | 1.2815+/-0.2733 | 1.8588+/-1.0955 | 1.3660+/-0.2732 | 1.6406+/-1.0110 | 1.5061+/-0.4900 | 1.4588+/-0.3855 |
| VGG19 | 1.4371+/-0.5419 | 3.0173+/-1.1538 | 2.6638+/-0.9600 | 2.4693+/-1.0423 | 2.6124+/-0.8336 | 3.0394+/-1.1825 | 1.1846+/-0.3475 | 1.7123+/-0.7078 | 1.6495+/-0.6638 | 1.7466+/-0.7488 | 1.8368+/-0.8803 | 1.6059+/-0.5379 |

**(D)**

| | BALD | Coreset | Least Confidence | Margin | Entropy | BADGE | BALD Pre | Coreset Pre | Least Confidence Pre | Margin Pre | Entropy Pre | BADGE Pre |
|---|---|---|---|---|---|---|---|---|---|---|---|---|
| RESNET18 | 1.0862+/-0.1284 | 1.1960+/-0.3401 | 1.2864+/-0.5337 | 1.2226+/-0.4378 | 1.3197+/-0.4724 | 1.3012+/-0.3916 | 1.0734+/-0.0642 | 1.1368+/-0.1436 | 1.0874+/-0.1685 | 1.1182+/-0.0996 | 1.0754+/-0.0978 | 1.1478+/-0.1570 |
| RESNET34 | 1.0970+/-0.1164 | 1.2613+/-0.3606 | 1.2600+/-0.3356 | 1.3000+/-0.3809 | 1.3023+/-0.3670 | 1.2795+/-0.4196 | 1.0782+/-0.2149 | 1.2415+/-0.4099 | 1.1346+/-0.4178 | 1.1693+/-0.3254 | 1.0040+/-0.2554 | 1.2246+/-0.4057 |
| RESNET50 | 1.0747+/-0.1670 | 1.3002+/-0.3334 | 1.2754+/-0.4127 | 1.3874+/-0.5029 | 1.2766+/-0.4983 | 1.3738+/-0.4258 | 1.0850+/-0.1683 | 1.0400+/-0.1865 | 1.1413+/-0.2171 | 1.0946+/-0.3873 | 1.0208+/-0.2279 | 1.1563+/-0.2988 |
| VGG11 | 1.0802+/-0.1073 | 1.1293+/-0.1990 | 1.4327+/-0.4196 | 1.3549+/-0.3775 | 1.3464+/-0.3963 | 1.3622+/-0.3473 | 1.1494+/-0.1379 | 1.2540+/-0.2744 | 1.3474+/-0.3945 | 1.2935+/-0.3685 | 1.2622+/-0.3307 | 1.3403+/-0.4190 |
| VGG16 | 1.0558+/-0.0797 | 1.2220+/-0.2639 | 1.2325+/-0.3048 | 1.3137+/-0.3140 | 1.2613+/-0.3291 | 1.2517+/-0.3108 | 1.0551+/-0.0840 | 1.1434+/-0.2396 | 1.1653+/-0.2667 | 1.2083+/-0.2440 | 1.1238+/-0.1998 | 1.1298+/-0.2742 |
| VGG19 | 1.0088+/-0.0916 | 1.1934+/-0.2374 | 1.1599+/-0.2488 | 1.1773+/-0.2282 | 1.0719+/-0.2438 | 1.1883+/-0.2642 | 0.9960+/-0.0471 | 1.1844+/-0.3158 | 1.0945+/-0.2633 | 1.1168+/-0.2398 | 1.0954+/-0.3066 | 1.1450+/-0.3259 |

Figure 27: Labeling efficiency over random acquisition, for all six acquisition function with and without unsupervised pretraining (represented with suffix *"pre"* after the name) along the columns across four datasets (A - CIFAR, B - FashionMNIST, C - MNIST, D - SVHN) and six DNN architectures along each row. Numerical values for labeling efficiency comparison for image dataset in Figure 1

## C.5.2 Text Dataset

**(A)**

| | BALD | Coreset | Least Confidence | Margin | Entropy | BADGE | BALD Pre | Coreset Pre | Least Confidence Pre | Margin Pre | Entropy Pre | BADGE Pre |
|---|---|---|---|---|---|---|---|---|---|---|---|---|
| BERT | 1.4543+/-0.6237 | 1.3781+/-0.5087 | 1.3392+/-0.6371 | 1.3019+/-0.5480 | 1.4502+/-0.5908 | 1.4557+/-0.7410 | 1.2072+/-0.2577 | 1.1103+/-0.2059 | 1.0747+/-0.1771 | 1.0406+/-0.3052 | 1.1103+/-0.2059 | 1.0917+/-0.1893 |
| ROBERTA | 1.2837+/-0.2388 | 1.2382+/-0.2371 | 1.1740+/-0.2194 | 1.2821+/-0.2963 | 1.2594+/-0.2067 | 1.1705+/-0.1944 | 1.5711+/-0.4081 | 1.7878+/-0.7104 | 1.5984+/-0.3382 | 1.9046+/-0.4553 | 1.7878+/-0.7104 | 1.2410+/-0.5323 |
| DISTILBERT | 1.6786+/-0.4551 | 1.8809+/-0.7263 | 1.6335+/-0.2919 | 1.9628+/-0.3397 | 1.2389+/-0.5131 | 1.1372+/-0.4297 | 1.2178+/-0.2865 | 1.0425+/-0.2673 | 1.2762+/-0.3893 | 1.1291+/-0.2470 | 1.0425+/-0.2673 | 1.2038+/-0.3108 |

**(B)**

| | BALD | Coreset | Least Confidence | Margin | Entropy | BADGE | BALD Pre | Coreset Pre | Least Confidence Pre | Margin Pre | Entropy Pre | BADGE Pre |
|---|---|---|---|---|---|---|---|---|---|---|---|---|
| BERT | 1.0490+/-0.0757 | 0.9515+/-0.0930 | 1.1459+/-0.0877 | 1.0936+/-0.0942 | 1.0600+/-0.1372 | 1.0026+/-0.1239 | 1.0069+/-0.0945 | 0.9318+/-0.1164 | 1.0739+/-0.0661 | 1.0898+/-0.0793 | 0.9318+/-0.1164 | 0.9030+/-0.0739 |
| ROBERTA | 1.0649+/-0.0521 | 0.9027+/-0.0167 | 0.9318+/-0.0544 | 0.9302+/-0.0949 | 0.9781+/-0.0709 | 0.8500+/-0.1012 | 1.3328+/-0.2039 | 1.0548+/-0.2282 | 1.4946+/-0.3441 | 1.2935+/-0.3711 | 1.0548+/-0.2282 | 1.1000+/-0.2881 |
| DISTILBERT | 1.3328+/-0.2039 | 1.0548+/-0.2282 | 1.4946+/-0.3441 | 1.2935+/-0.3711 | 1.0312+/-0.1070 | 1.1000+/-0.2881 | 1.0357+/-0.0304 | 0.8813+/-0.0752 | 0.9366+/-0.0668 | 0.9088+/-0.0525 | 0.8813+/-0.0752 | 0.9233+/-0.0580 |

**(C)**

| | BALD | Coreset | Least Confidence | Margin | Entropy | BADGE | BALD Pre | Coreset Pre | Least Confidence Pre | Margin Pre | Entropy Pre | BADGE Pre |
|---|---|---|---|---|---|---|---|---|---|---|---|---|
| BERT | 1.1394+/-0.3527 | 0.7678+/-0.1788 | 0.8882+/-0.1860 | 0.9167+/-0.1994 | 1.2587+/-0.2118 | 0.7688+/-0.2530 | 0.9869+/-0.3067 | 0.8029+/-0.1736 | 0.9379+/-0.2237 | 0.8826+/-0.3028 | 0.8029+/-0.1736 | 0.7853+/-0.1810 |
| ROBERTA | 1.0998+/-0.1519 | 0.8283+/-0.1009 | 0.9880+/-0.3052 | 0.8419+/-0.2488 | 1.0688+/-0.1619 | 0.7102+/-0.1072 | 1.2688+/-1.1650 | 0.7804+/-0.3056 | 1.2148+/-0.7437 | 0.7682+/-0.4591 | 0.7804+/-0.3056 | 0.4988+/-0.3306 |
| DISTILBERT | 1.2688+/-1.1650 | 0.7804+/-0.3056 | 1.2148+/-0.7437 | 0.7682+/-0.4591 | 1.1443+/-0.3200 | 0.4988+/-0.3306 | 1.1719+/-0.5499 | 0.8871+/-0.3330 | 0.9992+/-0.4967 | 1.0455+/-0.4604 | 0.8871+/-0.3330 | 0.9008+/-0.2555 |

**(D)**

| | BALD | Coreset | Least Confidence | Margin | Entropy | BADGE | BALD Pre | Coreset Pre | Least Confidence Pre | Margin Pre | Entropy Pre | BADGE Pre |
|---|---|---|---|---|---|---|---|---|---|---|---|---|
| BERT | 1.2256+/-0.2188 | 0.9754+/-0.1084 | 0.9754+/-0.1084 | 0.9754+/-0.1084 | 1.0271+/-0.2159 | 0.9128+/-0.1744 | 0.8170+/-0.1660 | 0.9444+/-0.1876 | 0.9444+/-0.1876 | 0.9444+/-0.1876 | 0.9444+/-0.1876 | 0.9537+/-0.1107 |
| ROBERTA | 0.6650+/-0.3159 | 0.7644+/-0.1301 | 0.9314+/-0.1453 | 0.9314+/-0.1453 | 0.8073+/-0.1455 | 0.8603+/-0.0897 | 1.1106+/-0.4716 | 0.8822+/-0.4748 | 0.8822+/-0.4748 | 0.8822+/-0.4748 | 0.8978+/-0.4844 | 0.9943+/-0.4501 |
| DISTILBERT | 1.0621+/-0.3017 | 0.7847+/-0.5615 | 0.7847+/-0.5615 | 0.7847+/-0.5615 | 1.0089+/-0.3376 | 0.8409+/-0.3462 | 2.0134+/-0.2789 | 2.1691+/-0.3483 | 1.9368+/-0.2767 | 1.9368+/-0.2767 | 2.1691+/-0.3483 | 1.8281+/-0.3384 |

Figure 28: Labeling efficiency over random acquisition, for all six acquisition function with and without unsupervised pretraining (represented with suffix *"pre"* after the name) along the columns across four datasets (A - AGNEWS, B - BANKS77, C - DBPEDIA, D - QNLI) and three DNN architectures along each row. Numerical values for labeling efficiency comparison for text dataset in Figure 3

# D    Focus 2: DNN Architecture Optimization during DAL

## D.1    Spread of performance across different acquisition function

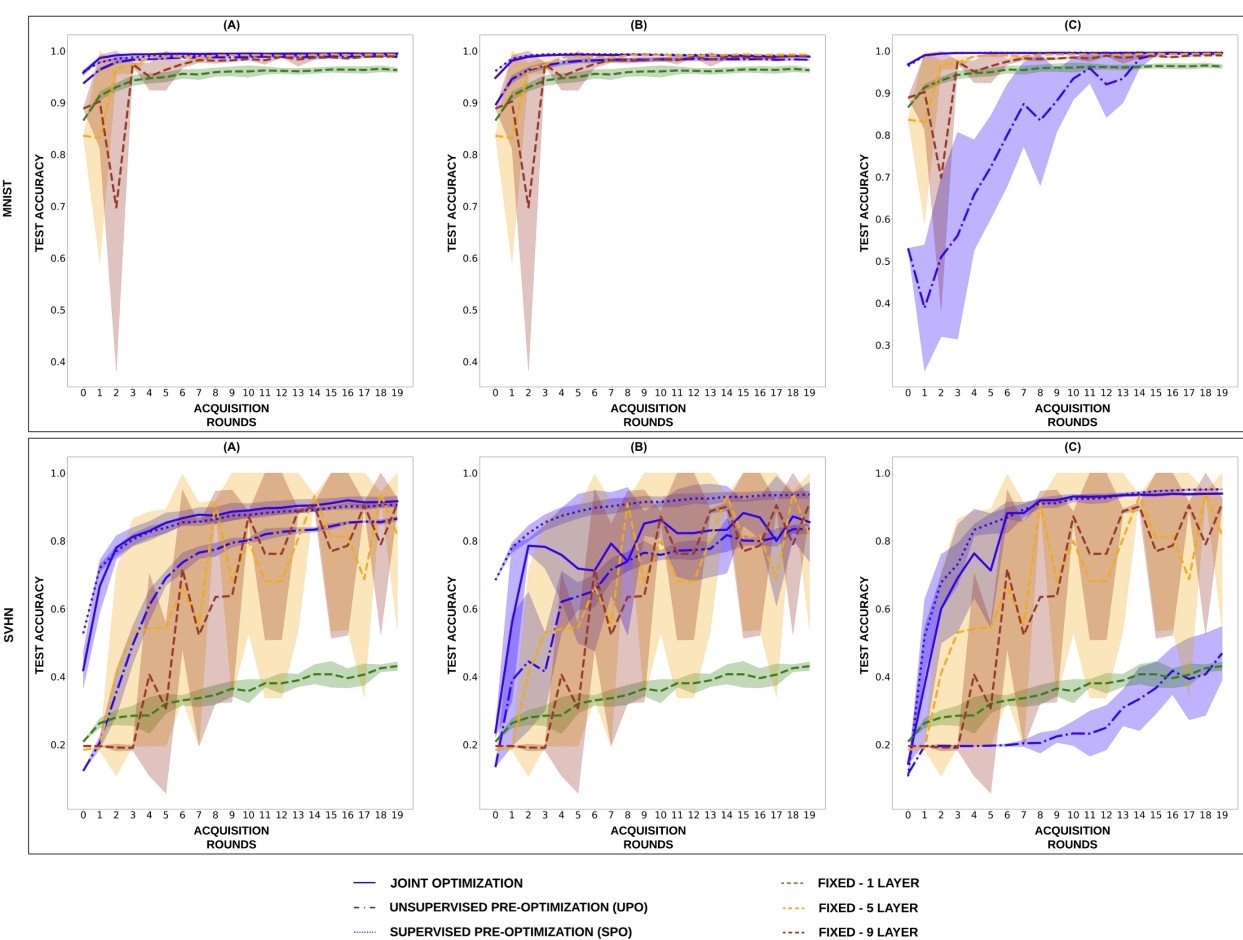

Figure 29: Spread of performance across different acquisition functions given pre-defined (fixed) CNN architecture, or jointly-optimized, supervised pre-optimized (SPO), unsupervised pre-optimized (UPO) using Depth-Dropout (A), BBDropout(B) and PDARTs(C) for MNIST (top row) and SVHN (bottom row) . Optimization of the DNN architecture, either jointly or during pre-training, in general improved over pre-defined CNN performance.

## D.2 Spread of performance across different network architectures

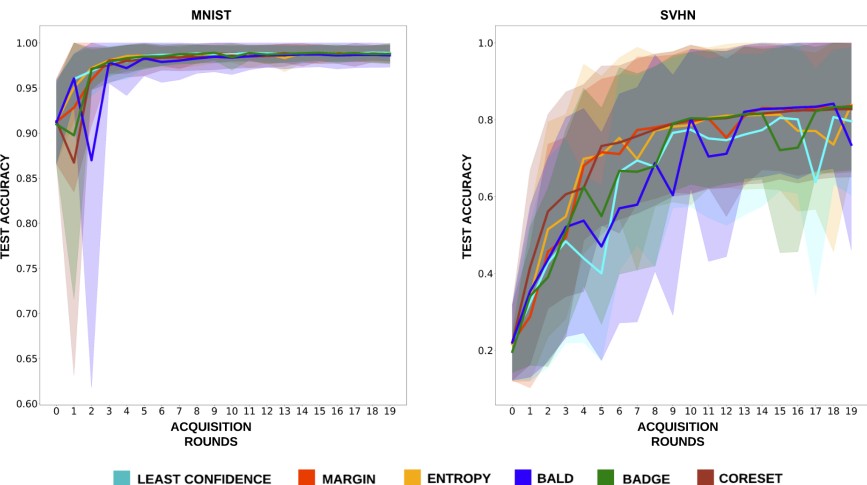

Figure 30: Spread of performance across different CNN architectures given acquisition functions for MNIST and SVHN, in the form of spread of test accuracy over the course of DAL.

