# OpenReview forum: "On the Interdependence between Data Selection and Architecture Optimization in Deep Active Learning"
_TMLR — Accepted by TMLR_

### Review · Reviewer_GpUt · 2024-09-25

**Summary Of Contributions:**

This paper investigates the problem of deep active learning (DAL) by empirically examining how deep neural network (DNN) architectures influence the feature space, and, consequently, the effectiveness of DAL. The study explores whether DAL performance is sensitive to model structures and investigates the impact of optimizing model architectures together with DAL on model performance gain. Extensive experiments are conducted using CNN and transformer models on both image and text datasets. The results demonstrate that DAL performance is highly dependent on the underlying model architecture. Intriguingly, the findings reveal that architecture search has a greater impact on DAL performance than the choice of acquisition function. This suggests that integrating architecture search with DAL could offer significant benefits for improving model performance.

**Audience:**

Yes

**Broader Impact Concerns:**

No broader impact concerns

**Claims And Evidence:**

Yes

**Requested Changes:**

**Clarification of notations**: Some notations, like $L$ and $U$, are introduced in Section 3.1 but not explained until Section 3.2. It would be helpful to mention that these will be explained later.

**Dataset Impact on Acquisition Function Performance**: From Fig. 3 and Fig. 7, it appears that the relative performance of acquisition functions may vary more depending on the dataset than on the model structure. It might be beneficial to discuss why dataset characteristics might have a larger impact on acquisition function effectiveness compared to model structures.

**Ranking Plot (Optional)**: While not necessary, it could be interesting to plot an average ranking of different acquisition functions with respect to model structures. Additionally, plotting the rank of acquisition functions with respect to datasets may provide further insights into how relative performance is affected by the nature of the dataset. This could add more depth to the comparison and help identify patterns in how acquisition functions behave across various settings.

**Strengths And Weaknesses:**

**Strengths**:

- Extensive experiments are conducted across different model structures and datasets.
- The finding that architecture search outweighs DAL acquisition strategies is potentially impactful and guides further DAL research.

**Weaknesses**:

- The paper does not provide any theoretical insight into how the decision boundary is affected by model structure. An analysis explaining the theoretical underpinnings of the impact of architecture on feature space and decision boundaries would have been beneficial.
- The conclusion that architecture search has more influence than acquisition functions is somewhat intuitive, given that a well-optimized architecture naturally improves learning efficiency.
- Other data formats (e.g., time series) are not tested, although it is foreseeable that this would not significantly affect the conclusions drawn in the paper.

---

> ### Author Response · Authors · 2024-10-31
> **Response to Reviewer GpUt**
>
> We thank the reviewer for your time and appreciation of the work, as well as your constructive feedback
>
> **Theoretical underpinning of the impact of architecture choices on decision boundaries**
> Please see the overall response for our revision in response to this comment, mainly in 1) tuning down our emphasis on this point in the main text due to our limited theoretical and empirical results supporting this point, and 2) adding experimental results in the Discussion and Appendix to show the impact of architectural choices on decision boundary in our experiments.
>
> **Intuitive conclusion regarding the effect of architectural choices**
> Indeed we acknowledge that this conclusion can be intuitive, although it was not consistently considered in existing DAL literature. As the main point of many existing DAL papers is to show that a new acquisition strategy outperforms existing strategies, we think it is important for the community to be aware that 1) the generalization of such conclusions needs to be cautiously checked against the choices of architectures and datasets considered in a paper, and 2) consistent experimental settings should be encouraged in the literature. We also hope to use this to spark new DAL research that considers how to best work with the choices and optimization of DNN architectures in the process of data acquisition.
>
> **Clarification of notations**
> We have added text in Section 3.1 in “DAL acquisition strategies” to explain notations like ‘L’ and ‘U’ as below:
>
>
> “*We consider the following acquisition functions, against random sampling as the baseline. For all below, we consider $L$ as the labeled datasets, $U$ as the unlabeled pool of dataset and $D$ as the complete dataset i.e. $D = L\ \cup\ U$. These are further explained in Section 3.2.*”
>
>
> **Dataset Impact on Acquisition Function Performance and Ranking Plot**
> Though not explicitly mentioned, different dataset differ from each other owing to the complexity of the dataset which includes underlying structure of dataset, dimensionality, noise, redundancy in data, decision boundary complexity, etc. These affect several factors that are important to the design of acquisition functions, including the uncertainty of a DNN that will affect uncertainty-based acquisition functions, the diversity of the data samples that will affect diversity-based acquisition functions, and the decision boundary which will affect all acquisition functions. Recent work [6] has shown that the ranking of acquisition functions vary depending on the nature of the dataset (balanced / imbalanced). Similarly, the scalability of acquisition functions has also been found to change with dataset [5]. Additionally, many existing works [1-4] in the active learning community have implicitly shown in their experiment that the performance of different acquisition functions differ for different datasets.  Our results presented in  Figure 1 to Figure 4 and Appendix C.1.1 confirmed this effect. Because the optimal DNN architecture depends on the underlying dataset, this probably also contributes to the observed dependence between the DAL performance and choices of DNN architecture.
>
>
> We thank the reviewer for this insight and have added the above discussion to Section 5.2 (Discussion).
>
> We also added the average ranking of acquisition functions for different datasets averaged over different networks in Appendix C.1.1. It shows that depending on the choice of dataset, the ranking of acquisition functions change. We referenced these figure in “Relative performance of acquisition functions” in Section 3.2 as below:
>
> “*This is confirmed by the relative average ranking of acquisition function (Appendix C.1.1) across different networks (RESNETs and VGGs for image data and BERT, ROBERTA, DISTILBERT for text data), showing that different acquisition functions have different benefits depending on the datasets*”
>
>
> **References**
> 1. Mayer, Christoph, and Radu Timofte. "Adversarial sampling for active learning." Proceedings of the IEEE/CVF Winter Conference on Applications of Computer Vision. 2020.
> 2. Beck, Nathan, et al. "Effective evaluation of deep active learning on image classification tasks." arXiv preprint arXiv:2106.15324 (2021).
> 3. Mittal, Sudhanshu, et al. "Parting with illusions about deep active learning." arXiv preprint arXiv:1912.05361 (2019).
> 4. Zhang, Jifan, et al. "Algorithm selection for deep active learning with imbalanced datasets." Advances in Neural Information Processing Systems 36 (2024).
> 5. Ji, Yilin, et al. "Randomness is the root of all evil: more reliable evaluation of deep active learning." Proceedings of the IEEE/CVF Winter Conference on Applications of Computer Vision. 2023.
> 6. Kim, Kwanyoung, et al. "Task-aware variational adversarial active learning." Proceedings of the IEEE/CVF conference on computer vision and pattern recognition. 2021.

---

### Review · Reviewer_x9aW · 2024-10-07

**Summary Of Contributions:**

This work provides an empirical study on the performance of six Deep Active Learning (DAL) methods under various settings, including different network architectures, datasets, initial labelled data size and subsequent acquisition size, and the presence/absence of data augmentation, pre-training, and network architecture optimisation methods. It demonstrates that effectiveness of these DAL methods do not necessarily generalise across different network architectures and training/valuation datasets. Further, data augmentation and network optimisation (both joint and supervised prior methods) are generally beneficial when paired with DAL methods. In contrast, unsupervised pre-training and unsupervised network architecture optimisation are found less useful and sometimes even harmful to DAL methods.

**Audience:**

Yes

**Broader Impact Concerns:**

Nil.

**Claims And Evidence:**

No

**Requested Changes:**

Critical:

Please address the weaknesses W. 1 and W. 2.

Mild:

Please address the weakness W. 3.1 - W. 3.4.

In Figure.~1, it is interesting that DAL method selects data around the decision boundaries in both input and feature spaces at early stages. But at a late stage (last row), it seems to acquire data that are farther away from both decision boundaries, which is most noticeable with model X-4-2-y. Is there an explanation for such a phenomenon?

In Figure.~1, is it possible to disclose which DAL method is used here? Additionally, are the decision boundaries obtained before or after training on the newly acquired data on the same row?

Trivial:

Typo in Page~5 - last paragraph: ``Entroy" -> Entropy.

Acronym ``IBP" undefined in Page. 13 above Equation. 9.

**Strengths And Weaknesses:**

Strengths

S1 - This work has performed an extensive empirical investigation on several existing DAL methods under various settings, revealing that their effectiveness do not necessarily generalise across different settings.

S2 - It has demonstrated that data augmentation and supervised- prior and joint network architecture optimisation are potentially beneficial in making DAL methods perform more consistently across different settings.

Weaknesses

W1 - Over-claim the contribution: In abstract, the work claims to shed light on the impact of changes in feature space on optimal data selection. However, the only content pertaining to the feature space is Figure.~1 and its description in ``Introduction" section which only demonstrates that feature space corresponding to three small-scale networks are different. It does not establish a causal relationship between feature space changes and the data acquisition of DAL methods.

W2 - Over-generalisation: In Section.~5 - 1st paragraph: this work suggests network architecture optimisation as a potential solution to ensuring the generalisation of DAL methods. However, the beneficial effects of joint/prior supervised network architecture optimisation on DAL methods are only observed for a single type of network with various depths. Shouldn't the work also be cautious in suggesting its generalisation just like it cautioning in overly generalising DAL methods across different network architectures?

W3 - Presentation:

W3.1 - In Focus 2, it is unclear what ``DNN architectures with 1, 5, and 9 layers" are. It is also unknown which DNN architecture (number of layers) is used as baseline for the network architecture optimisation experiments.

W3.2 - Figure. 10 and Figure. 21 are associated with caption "Spread of performance across different architectures". However, the plot only illustrates how test accuracy changes with increasing data acquisition rounds using six DAL methods on CIFAR and FASHIONMNIST datasets. Also, the shaded areas of different DAL methods are highly overlapped, it is difficult to tell the test accuracy ``spread" of individual methods.

W3.3 - Figure.~20 is associated with caption ``Spread of performance across different acquisition functions". However, none of the DAL method is included.

W3.4 - Figure.~15 - 19, the ``spread" for red and blue plots are missing in all VGG models (second row).

---

> ### Author Response · Authors · 2024-10-31
> **Response to Reviewer x9aW**
>
> We thank the reviewer for your time and appreciation of the work, as well as your constructive feedback
>
> **W.1 Over-claim the contribution**
> Please see our combined response at the top. To summarize, we have made changes to our title, abstract and introduction in reflection to the constructive suggestion provided. We have also added new results that show decision boundary changes with change in networks (RESNETs and VGGs) for image dataset (CIFAR10, FashionMNSIT, MNIST and SVHN) to Section 5.1. And Appendix C.2
>
>
> **W.2 Over-generalisation**
> We agree with the reviewer that the observations are based on single type of network structure i.e. CNN networks. We showed the experiments based on said network type due to its wide use. We agree that a broader class of network structures would provide a more detailed view which we have mentioned in the 1st paragraph of Section 5.3 and further stated that to generalize it better we would need to extend it to larger overparameterized architectures as below.
>
> “*Investigations in Focus 2 of the current study is focused on relatively small CNN based architecture optimization. To further generalize the findings, future studies need to extend to larger overparameterized architectures such as RESNET, VGG, and transformers. Such extension will provide a more comprehensive understanding of how optimization of different network architectures influence data optimization, providing insights into the applicability of joint architecture and data optimization strategies across a variety of DNN types.*”
>
>
> **W3.1  Clarification on DNN architectures with 1, 5, and 9 layers**
> We would like to clarify that, by DNN architecture with 1, 5 and 9 layers, we are stating that these are DNN architectures with 1, 5 and 9 convolutional layers from start to end. We are using these networks with fixed sizes as baselines to compare with networks with optimized architectures. We have added text in “The effect of optimized DNN architectures” in Section 4.2 to clarify this as below:
>
> “*Fig. 7 summarizes the DAL performance across different acquisition functions when applied to architecture optimization – Depth-Dropout (A), BBDropout (B), and PDARTs (C) – and fixed DNN architectures with 1, 5, and 9 convolutional layers (between input and output layers of the network) on CIFAR10 and FashionMNIST. The fixed architectures are used as baselines to compare with the networks with optimized architectures.*”
>
>
> **W3.2 and W3.3 Clarification on “Spread of performance”**
> The “spread of performance across different architectures” refers to the spread of test-accuracy obtained across different architectures using a fixed acquisition function (AF). Similarly, “the spread of performance across AFs” refers to the spread of test-accuracy obtained across different AFs (described in section 3.1) using the same underlying architecture type. The former is shown in figure 10 and 21 (Now 9 and 30), where this spread of test-accuracy is represented by the shaded (lowest test-accuracy to high test-accuracy) and the latter is shown in figure 8 and 20 (Now 7 and 29). Indeed as pointed out by the reviewer, the spread induced by different architectures (in Fig 9 and 31) is much larger in comparison to that induced by different acquisition functions (in Fig 7 and 30). In other words, even with the performance spread induced by  different AFs, different architecture still can be ranked by their performance; in comparison, with the performance spread included by different architectures,  the performance of different AFs become highly overlapped and indistinguishable as noted by the reviewer. This is exactly the basis on which we draw the conclusion that architecture optimization outweighs the choice of AFs in the paper.
>
>
> **W3.4 Correction of figures**
> We have updated the figure to reflect the “spread” for all VGG models in Figure 15 to 19 (Now 22 - 26) of the paper.
>
> **Clarification regarding Figure~1**
> We used entropy as the active learning method to select the data in Figure 1 (Now Figure 12 in Section 5.1) where the plots are shown after training on acquired data. In earlier stages, the model is uncertain about the data near the boundary; as a result the selection is concentrated there. The acquired data is then added to the training data and the model is retrained. As the training progresses, the model is more certain about the boundary and thus selects the data away from the boundary
>
> **Other Comments**
> The typo in page~5 has been fixed from “Entroy” to “Entropy” and the acronym IBP has been provided as “Indian Buffet Process (IBP)”

---

### Review · Reviewer_Htho · 2024-10-23

**Summary Of Contributions:**

The authors empirically investigate the role of Deep Neural Network (DNN) architectures in Deep Active Learning (DAL), addressing two complementary questions:

1. Whether any single DAL acquisition strategy consistently outperforms others across various architectures and datasets. Through their experiments, they demonstrate that among the diverse schemes tested, no single strategy consistently prevails.
2.  Whether optimizing the architecture can provide greater benefits than the improvements achieved through DAL itself, which they answer affirmatively.

While the questions posed are intriguing, there are some concerns with the claims of the study, as outlined below.

**Audience:**

Yes

**Broader Impact Concerns:**

Not applicable.

**Claims And Evidence:**

Yes

**Requested Changes:**

### Major
- Focus 1:
    - The error bars reported in Figures 1-4 appear to be quite large, and I assume these are calculated over three seeds. Increasing the number of seeds would likely make these findings more reliable, as the current error bars suggest that all methods perform statistically similarly. This may result in qualitative conclusions (such as those in the section “Relative performance of acquisition functions”) being based on visual differences in the bars rather than the error bars. Additionally, I have two minor suggestions:
        - Consider using a log scale for the y-axis in the plots describing label efficiency. This could better represent the differences visually.
        - Reporting the exact values in a table would also be helpful. Specifically, what is the random test accuracy? At what iso-accuracy levels are the labeling efficiencies reported?
    - Regarding the conclusion on data augmentation in Figure 6, it seems that the models trained with data augmentation simply see more data points, which could explain the observed increase in accuracy. For a fair comparison, the x-axis should reflect the same number of data points seen across models.
- Focus 2:
    - One of the key issues with the conclusions drawn from this part (e.g., in Figure 8) is that the error bars represent the spread across different data acquisition strategies. To measure performance spread due to architecture optimization, a more appropriate experiment would compare each data acquisition strategy individually, alongside random sampling, while keeping the architecture either fixed or optimized for each strategy. The current experiments showing that architecture optimization has a stronger effect might be a case of “Simpson’s paradox.” Either the data for each acquisition strategy should be provided, or the claim should be rephrased to reflect the metric being evaluated (e.g., “averaging across data acquisition strategies, optimizing model architecture provides higher gains”).
    - Additionally, as shown in Figure 8, supervised pre-optimization (SPO) performs comparably to joint optimization. The fact that using initially labeled data to optimize architecture before DAL performs competitively suggests an alternative interpretation: data acquisition from an optimized network can be as competitive as joint optimization. Although this point is mentioned in the paper, the main conclusions seem to overlook its significance.

### Minor
- Equation mistakes in DAL acquisition strategies described on Pg 5:
    - The equation for margin sampling should use argmin instead of argmax to be consistent with the notation.
    - The definition of entropy sampling is missing a minus sign in the entropy formula.
    - Symbols used in the BALD sampling equation are not defined.
- “Entroy” should be corrected to “Entropy” on Page 5.
- Clarification on Evaluation Metrics for Part 1, Pg 6
    - Does the second metric in Figures 3 and 4 represent an average over all acquisition rounds when it measures “the percentage of gain in test accuracy over random acquisition at each acquisition round”? This should be clarified in the “evaluation metrics” section.
- Typographical error in Page 9 in “Effect of data augmentation” section: “1) most acquisition functions tested exhibited either a substantially larger gain over the random acquisition (e.g., CIFAR10 for all networks, MNIST for RESNETs and VGG11), or a reduced error in the gain (CIFAR10, SVHN for all networks)”. This is contradictory statement e.g. wrt CIFAR10.
- Pg 14, a full stop is missing after “M” in the phrase “each layer is limited to M.”

**Strengths And Weaknesses:**

### Strengths
- The problem is well-formulated and the past literature is summarized well.
- The authors conduct extensive empirical experimentations throughout the paper, such as all results are presented across various datasets, model architectures, model optimization schemes, data augmentation and data acquisition schemes as applicable.
- The inclusion of multiple approaches to architecture optimization, particularly supervised and unsupervised pretraining, adds an interesting dimension to the discussion of DAL and opens new avenues for future research.

### Weaknesses
 Summary of weaknesses is provided, for details please look at requested changes.
- I think many of the conclusions here are qualified for small networks working with small data given the size of the empirical experiments. It is unclear if similar findings hold for modern over-parametrized networks. Though this is not an issue, I think it should be specifically mentioned in the paper.
- Some of the conclusions are not thoroughly backed by the experimental evidence.
- Given the suggested future work of setting up a benchmark, it would significantly strengthen the paper if they could release the code for their experiments. This would enhance reproducibility and allow the community to build upon their findings.
- There are minor errors throughout the paper, indicating the need for a more thorough review and proofreading.

---

> ### Author Response · Authors · 2024-10-31
> **Response to Reviewer Htho**
>
> We thank the reviewer for your time and appreciation of the work, as well as your constructive feedback
>
> ### **Extension to modern networks and larger datasets**
> We agree with the reviewer that the current work does not include larger networks. We agree that a broader class of modern network structures, which are larger and overparameterized, would provide a more detailed view. We have acknowledged these in the 1st paragraph of Section 5.3 as below.
>
> “*Investigations in Focus 2 of the current study is focused on relatively small CNN based architecture optimization. To further generalize the findings, future studies need to extend to larger overparameterized architectures such as RESNET, VGG, and transformers. Such extension will provide a more comprehensive understanding of how optimization of different network architectures influence data optimization, providing insights into the applicability of joint architecture and data optimization strategies across a variety of DNN types.*”
>
> We also agree that the current findings are based on smaller datasets and future work needs extension to larger and more complex datasets. We have acknowledged these in Section 5.3 as below:
>
> “*Additionally, current work is focused on relatively smaller datasets like MNIST, FashionMNIST, SVHN and CIFAR10 and broadening the scope of datasets to include larger datasets like CIFAR100 (Krizhevsky, 2009a), Imagenet (Deng et al., 2009), CelebA (Liu et al., 2015), etc will help generalizing the observations across more complex datasets space.*”
>
>
> ### **Conclusion not thoroughly backed by experiments**
> Based on comments from both this and other reviewers, we have revised our conclusion (or added new results) to address this concern. Primarily:
> 1. Please see the overall response regarding our revision to tune down our motivation/conclusion on the effect of decision boundary changes on data acquisition.
> 2. We have revised our conclusion about Focus 2 to acknowledge that the observations are made based on only CNN networks (revised in the 1st paragraph of Section 5.3).
> 3. For other relevant revisions related to our stated conclusions, please see the following response in detail.
>
> ### **Code Release**
> We will release our codebase with the final version of the manuscript.
>
> ### **Responses related to Focus 1**
> **Response related to additional seeds in Focus 1**
> As the reviewer constructively stated, increasing the number of seeds would be desirable. Currently, the results we report in the paper are based on three seeds. We are currently running more seeds and will be happy to update our results for the final version of the manuscript.
>
> **Log plot for label efficiency in Focus 1**
> We would like to thank the reviewer for taking time to study the results in such depth and for the suggestion. In figure 2 and 4, that show the labeling efficiency, there is a large difference in range of values of the bars and the errors. As a result, the log plots were difficult to interpret as the height of the bar became unobservable in comparison to the now larger error bar. Due to these reasons, we have decided to show the results in the current scale.
>
> **Tabular representation of labeling efficiency**
> We have added numerical values of labeling efficiency for Figure 2 and 4 (Now Figure 1 and 3) in Appendix~C.5. We reference this in section 3.2 as below:
>
> “*The numerical values for Figure 1 and Figure 3 can also be found in Appendix C.5*”
>
> Regarding the reported labeling efficiency shown in Figure 1 and 3, we report the mean labeling efficiency across the acquisition rounds.
>
> **Regeneration of Figure 6 to reflect same number of data along x-axis**
> To clarify, Figure 6 (Now 5) shows the effect of increasing the initial labeled size. We do agree with the reviewer and in response to the suggestion we have updated Figure 6 (Now 5) as well as Figures 11-14 (Now 18-21) in Appendix C.3 so that the x-axis reflects the same number of labeled data points seen across models.
>
> As the reviewer pointed out, the observed increase in accuracy is due to the increase in the initial number of labeled data. The increase in initial label data size effectively shifts the performance curve to the right compared to the lower initial label data. To clarify this better, we have added extra text in section “Effect of initialization and acquisition size” in Section 3.2 as well as caption of Figure 6 as below
>
> “*This shows that increasing the initial labeled data size effectively shifts the performance curve to the right compared to a smaller initial labeled data size.*”

---

> > ### Author Response · Authors · 2024-10-31
> > **Response to Reviewer Htho continued**
> >
> > ### **Responses related to Focus 2**
> >
> > **Clarification on conclusions drawn from Focus 2**
> > To clarify, the shade in Figure 8 (Now Figure 7) shows the spread of performance exhibited by different acquisition functions (described in section 3.1) including random acquisition functions for each type of network (optimized and fixed) considered. Similarly, the shade in Figure 10 (Now Figure 9) shows similar spreads exhibited by different networks for each type of acquisition function. We can see from Figure 8 that the gap or spread of different acquisition functions (shown with shaded region) for optimized networks is very small for all data types. In contrast, when we look at Figure 10, for a given acquisition function, when we consider different types of network, the spread of performance exhibited is far wider. This led to the conclusion that network optimization is equally if not more important during DAL. This relation between data acquisition and network optimization is studied for different networks, acquisition functions, and datasets which should reduce the chances of “Simpson’s paradox”.
> >
> > **Inclusion of competitive nature of SPO and joint optimization in Conclusion**
> > As the reviewer has pointed out, we have mentioned in the discussion that optimization of the network either preceding DAL or jointly with DAL plays an important part in DAL with supervised pre-optimization potentially being more beneficial. We conclude this fact in the paper with following text
> >
> > “*We show that the choices of DNN architecture substantially influence and outweigh data optimization in DAL, and that its optimization helps increase the benefits of active data selection.*”
> >
> > To ensure, the message is clearly conveyed, thanks to the reviewers suggestion, we have added following text after the above line
> >
> > “*, with supervised pre-optimization being most beneficial followed by joint optimization.*”
> >
> > ### **Minor changes**
> > **Clarification regarding “Effect of data augmentation”**
> > We would like to clarify that we were trying to convey by adding data augmentation, the gain (gap) between random acquisition and other acquisition was increased (height of the bar graph in figure 7 (Now 6)). On the other hand, it also showed reduced error in gain i.e. the error across different seeds for acquisition functions reduced in comparison to without augmentation. For CIFAR10 both were observed but for other datasets either of the two was observable. We have changed the text to clarify this as follows:
> >
> > “*1) most acquisition functions tested exhibited either a substantially larger gain over the random acquisition (\textit{e.g.}, MNIST for RESNETs and VGG11), or a reduced error in the gain (SVHN for all networks). CIFAR10 saw both effects across all networks.*”
> >
> > **Clarification on “Evaluation Metrics”**
> > Yes, the metric is evaluated as average over acquisition rounds. We have modified the text in the “Evaluation metrics” section in 3.2 to clarify this as below
> >
> > “*a new metric that measures the percentage of gain in test-accuracy over random acquisition at each acquisition round averaged over all acquisition rounds*”
> >
> > **Equation Correction**
> > We have corrected the equation for “Entropy” and “Margin Sampling” in Section 3.2. Additionally, the symbol used in the BALD equation has been mentioned.
> >
> > **Typographic corrections**
> > A full stop is missing after “M” in the phrase “each layer is limited to M.” has been added.

---

> > > ### Comment · Reviewer_Htho · 2024-11-18
> > >
> > > Thank you for the detailed answers and changes. I look forward to better error bars with more seeds in the comparison for Focus 1.  I am still not clear on following point and would recommend clarification (or even removal of the point from main paper).
> > >
> > > ### On "Regeneration of Figure 6 (now Figure 5) to reflect same number of data along x-axis"
> > > - I am still not convinced with Figure 5 after changes. To me, Figure 5 (and corresponding figures in appendix) do not clearly show the benefits of larger initialization size. Looking at the plots, I don't convincingly see solid lines being consistently higher, giving higher test accuracies, at same amount of labeled data. In-fact for red and green curves opposite seems to be true --> solid lines are consistently below dashed lines => poorer test-set performance for experiments with higher initialization size at iso-data seen by the models. Therefore, following statement is wrong and should be clarified:
> > > > "The increase in initial pool size brought a gain in test performance for all
> > > DNNs, although this performance gain decreased for all acquisition functions as DAL proceeds."
> > >
> > > - Moreover, the line following above in caption is contradictory:
> > > > "This shows that increasing the initial labeled data size effectively shifts the performance curve to the right compared to a smaller initial labeled data size."
> > >
> > > Shifting towards the right implies the method needs more labeled data for iso-test accuracy, and hence is worse. Therefore, this line which follows plots more closely seems to suggest that increasing the initial labeled size is resulting in worse-performance, contradicting the main conclusion of this plot and section.
> > >
> > > ### Aside:
> > > Looking at comments from other reviewers, I think it will be interesting for authors to compare their findings against Ref 1 which studies pool-based active learning theoretically as well as empirically. It also shows how the chosen DAL methods in this study, which typically exploit most-uncertain points, can be sub-optimal given various properties of the chosen problem such as dataset, model sizes relative to dataset size, SNR in the dataset etc.
> > >
> > > [1] Kolossov, G. et al., "Towards a statistical theory of data selection under weak supervision." In The Twelfth International Conference on Learning Representations.
> > >
> > > I have no further suggestions.

---

> > > > ### Author Response · Authors · 2024-11-22
> > > > **Response to Reviewer Htho**
> > > >
> > > > We thank the reviewer for your time and detailed recommendations on the work.
> > > >
> > > > **Effect of initialization size on active learning**
> > > >
> > > > We agree with the reviewer that the increased initialization size doesn’t offer benefit in comparison to smaller. To reflect this, we changed the text on “Effect of initialization and acquisition size” section as follows:
> > > >
> > > > *As shown in Figure 5, the increase in the initial labeled data size did not show a benefit compared to the smaller initialization size. Increasing the initial labeled data size effectively shifts the performance curve to the right compared to a smaller initial labeled data size. We observe that the performance of BADGE, Entropy, and Margin was worse compared to a lower initialization size at the same number of labeled data while that of BALD, and Coreset were unchanged.*
> > > >
> > > > We also shorted the caption in Figure 5 to
> > > >
> > > > *Comparison of test accuracy of different acquisition functions for different DNNs trained on FashionMNIST with an initialization size of 1000 (dashed) vs. 5000 (solid). Similar results on other datasets can be found in the Appendix C.3.*
> > > >
> > > > **Comparing findings with suggested paper**
> > > >
> > > > We thank the reviewer for the recommendation. As stated by the reviewer and in the paper, we agree that uncertainty based sampling is not always optimal and there should be a balance between that and data properties (diversity based methods). Due to this, we also included hybrid methods like BADGE and a diversity based method as Coreset.
> > > >
> > > > Furthermore, in regards to choice of model with active learning, we found that blindly increasing the model complexity doesn’t always help but rather can hurt the performance. Thus, there should be an investigation on choice of model and if possible some sort of optimization of the architecture. This was also observed in the suggested paper which states “better surrogate models do not always lead to better data selection”. This supports our claim that the optimized network architecture is as important if not more during data selection because a “better surrogate” in one acquisition step might not be best for the next.
> > > >
> > > > At the moment, we added following text in the start of our discussion, in section 5.1 to start our discussion on which adds to the Relation between DAL acquisition, DNN architecture and decision boundaries:
> > > > *What may explain the observed interdependence between the DNN architecture and data acquisition? \citep{kolossov2023towards} showed in their work that a better performing model is not always an ideal choice during active learning.*
> > > >
> > > > We will review the paper in further detail for a more thorough comparison between our findings and that presented in this reference.

---

### Author Response · Authors · 2024-10-31
**Combined Response**

We would like to thank the reviewers for their constructive suggestions to improve the quality of this work. Below we summarize major changes made in response to the common suggestions made by the reviewers, and leave detailed revisions addressing each reviewer’s comments in the individual responses.

**Limited theoretical support and result/discussion regarding optimal decision boundary and feature spaces**
We presented this hypothesis based on our observations that the change in model architecture affects the decision boundary, as illustrated in Fig. 1 (Now figure 12) of the manuscript. This was also shown in other works: in [2], the decision boundary of a fixed architecture was shown to continuously change even in the later phases of training. This variability was shown to inversely affect generalizability and reproducibility of the model [3]. When the network was changed, the decision boundary was also shown to visibly change [1]. Though intuitive, this effect and its relation with data acquisition design was often overlooked in DAL works, which motivated us to present our original submission through this lens.

However, we do acknowledge that our main results provided limited theoretical or empirical results to delve deeper into this relation among DNN architecture, optimal decision boundary, and the associated optimal data acquisition strategies. In reflection of this, we have made the following changes to the revised manuscript.


- **Title change**: We have changed our manuscript’s title to “On the Interdependence between Data Selection and Architecture Optimization in Deep Active Learning”, to focus more on the relation between these DNN architecture and data acquisition which constitutes the main results of our paper. This tunes down the emphasis on the relation between the changes in decision boundary and optimal data acquisitions.
- **Change in Introduction**: We have accordingly revised the abstract and Introduction to motivate and rationalize more from the lens of dependence between data and architecture optimization, and removed discussion regarding the change in optimal decision boundary and its relation to data acquisition. We also moved Figure 1 and its related discussion partly to Section 5.1 (Discussion) and partly to Appendix (see below)
- **Addition of discussion and results**: We moved discussion related to the change in decision boundary as a result of DNN architectures and its relation to optimal data acquisition in Section 5.1 (Discussion). In addition, in Section 5.1 and Appendix C.2 we also added results – leveraging the visualization method presented in [1] – to visually demonstrate the change in decision boundary as a result of the DNN architecture in RESNETs (18,34,50) and VGGs (11,16,19) on image datasets (CIFAR10, FashionMNIST, MNIST and SVHN). We used these to discuss the change in decision boundary as a potential mechanism through which DNN architecture affects optimal data acquisition, via which to bring out the needs of future theoretical works on this topic.


**References**
1. Somepalli, Gowthami, et al. "Can neural nets learn the same model twice? investigating reproducibility and double descent from the decision boundary perspective." Proceedings of the IEEE/CVF Conference on Computer Vision and Pattern Recognition. 2022.
2. Mickisch, David, et al. "Understanding the decision boundary of deep neural networks: An empirical study." arXiv preprint arXiv:2002.01810 (2020).
3. Lei, Shiye, et al. "Understanding deep learning via decision boundary." IEEE Transactions on Neural Networks and Learning Systems (2023).

---

### Decision · Action_Editor_8rtQ · 2024-11-27

**Recommendation:** Accept as is

**Comment:**

All three reviewers agree that the paper should be accepted and published in TMLR. I share their opinion and am therefore recommending acceptance.

**Audience:**

Deep active learning and how to efficiently deal with limited labels is an ongoing research area of great importance. As such it will be of interest to part of the TMLR audience.

**Claims And Evidence:**

The paper evaluates various acquisition strategies for neural-network-based active learning with multiple architectures on various data sets.

The paper is primarily empirical and lacks a theoretical foundation. Nevertheless, its broad structure provides a clear and convincing overview, despite minor possible improvements, e.g., the number of seeds is rather limited.